# 🦖✉TikZilla: Scaling Text-to-TikZ with High-Quality Data and Reinforcement Learning

**Christian Greisinger & Steffen Eger**

University of Technology Nuremberg (UTN)

{christian.greisinger,steffen.eger}@utn.de

## Abstract

Large language models (LLMs) are increasingly used to assist scientists across diverse workflows. A key challenge is generating high-quality figures from textual descriptions, often represented as TikZ programs that can be rendered as scientific images. Prior research has proposed a variety of datasets and modeling approaches for this task. However, existing datasets for Text-to-TikZ are too small and noisy to capture the complexity of TikZ, causing mismatches between text and rendered figures. Moreover, prior approaches rely solely on supervised fine-tuning (SFT), which does not expose the model to the rendered semantics of the figure, often resulting in errors such as looping, irrelevant content, and incorrect spatial relations. To address these issues, we construct DaTikZ-V4, a dataset more than four times larger and substantially higher in quality than DaTikZ-V3, enriched with LLM-generated figure descriptions. Using this dataset, we train TikZilla, a family of small open-source Qwen models (3B and 8B) with a two-stage pipeline of SFT followed by reinforcement learning (RL). For RL, we leverage an image encoder trained via inverse graphics to provide semantically faithful reward signals. Extensive human evaluations with over 1,000 judgments show that TikZilla improves by 1.5-2 points over its base models on a 5-point scale, surpasses GPT-4o by 0.5 points, and matches GPT-5 in the image-based evaluation, while operating at much smaller model sizes. Models and datasets are available on https://huggingface.co/collections/nllg/tikzilla.

## 1 Introduction

Large language models (LLMs) have become an increasingly valuable tool for scientists across domains (Bi et al., 2024; Eger et al., 2025), driven by scaling model size, hardware, and data (Minaee et al., 2025), as well as by research expanding multimodal capabilities (Wu et al., 2023) and enabling advanced reasoning (Lu et al., 2024). As a result, an increasing number of tools have been developed to support scientists throughout the research process, which range from idea generation (Gottweis et al., 2025) to the full automation of scientific outputs (Lu et al., 2024). However, these fully autonomous tools are still far from meeting the high scientific standards required for practical use. Achieving such standards involves overcoming complex subtasks, such as generating accurate scientific images based on textual descriptions (Rodriguez et al., 2023; 2024; Zou et al., 2024).

Graphics programming languages such as TikZ are the de facto standard in academia due to their precision, interpretability and seamless integration in the LaTeX ecosystem. However, their steep learning curve and highly varied syntax make them difficult for both humans and LLMs to master (Belouadi et al., 2024a). Prior works have attempted to bridge this gap by finetuning LLMs on caption-TikZ pairs (Belouadi et al., 2024a; 2025). Due to the sparsely available data, Belouadi et al. (2025) leverage captioned images without the underlying graphics program available, therefore having access to a much richer dataset. However, these efforts remain limited by noisy captions, a lack of executable and standardized TikZ code, as well as a lack of direct visual feedback, leaving models prone to low compilation rates, hallucinations, overly long responses, and low-quality outputs.

We address these limitations by constructing DaTikZ-V4, a dataset more than 1.5M instances larger than its predecessor, sourced from arXiv, GitHub, TeX StackExchange (TeX SE), and synthetic data.

Table 1: Exemplary scientific TikZ figures produced by one baseline LLM (GPT-4o) and two of our finetuned LLMs (TikZilla-3B and TikZilla-3B-RL) using the prompts from the first column which have been VLM augmented based on the Ground Truth figures in the second column. ■-boxed figures have been rated as very good, ■ as good, ■ as bad, and ■ as very bad by human annotators. Additional examples are provided in the Appendix (Table 10, 11, 12, and 13)

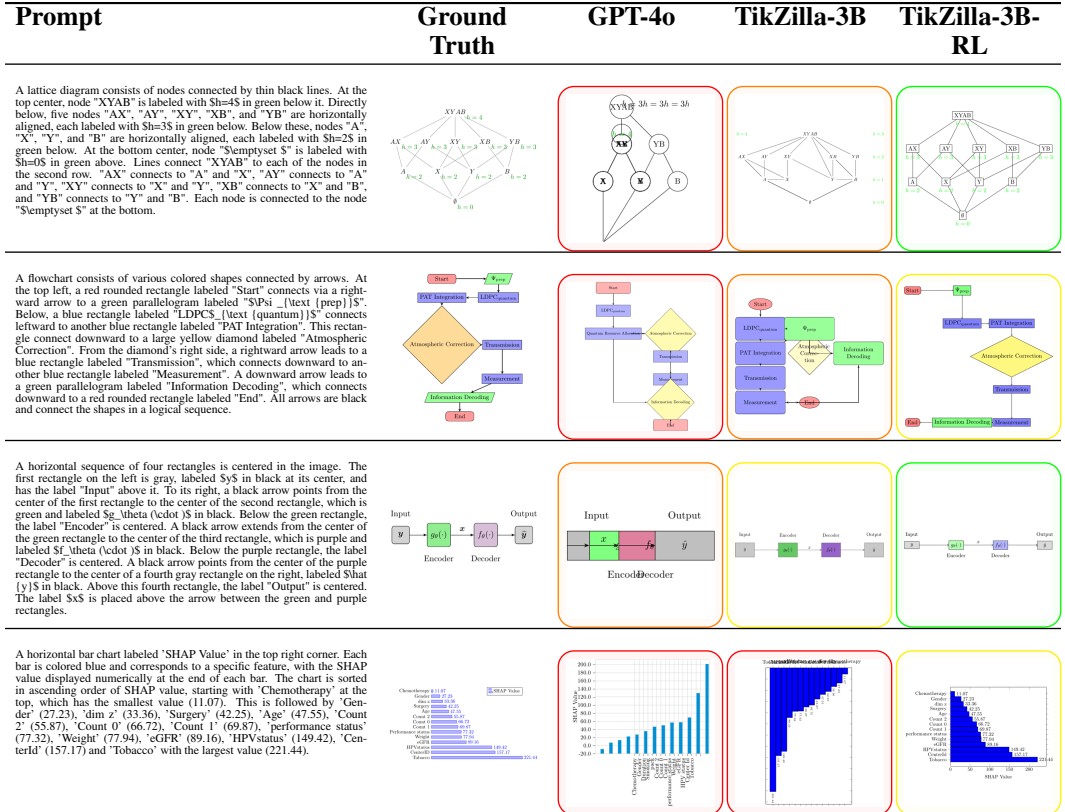

To improve data quality, we introduce an LLM-based debugging pipeline that repairs uncompilable TikZ code, and employ Vision Language Models (VLMs) to generate accurate figure descriptions. Building on DaTikZ-V4, we develop TikZilla, a family of small Qwen-based models (3B and 8B) trained with a two-stage pipeline: Supervised Finetuning (SFT) for syntax alignment, followed by Reinforcement Learning (RL) with a reward model trained on the Image-to-TikZ task beforehand. We find that this approach substantially improves Text-to-TikZ generation quality, where even models as small as 3B parameters outperform GPT-4o across automatic metrics and over 1,000 human judgments spanning four baseline LLMs. Table 1 shows examples with corresponding human ratings. We summarize our key contributions as follows:

- **Caption Quality Analyisis:** We show that widely available captions are insufficient for reconstructing figures.

- **Scaling Dataset Size:** We introduce DaTikZ-V4 with over 2M unique TikZ samples, sourced from newer arXiv submissions and GitHub, quadrupling the scale of prior datasets.

- **Data Quality Enhancements:** We combine (1) improved rule-based filtering (e.g., dynamic package inclusion), (2) VLM-based scientific figure descriptions, and (3) an LLM debugging pipeline for uncompilable TikZ code.

- **Reward Model:** We finetune an image encoder on the Image-TikZ task using our larger TikZ corpus, providing more semantically meaningful rewards for RL optimization.

- **TikZilla Models:** We release TikZilla, a family of small open-source Qwen models (3B and 8B). TikZilla outperforms GPT-4o across automatic and human evaluation, and matches GPT-5 in image-based evaluation, despite operating at much smaller model sizes.

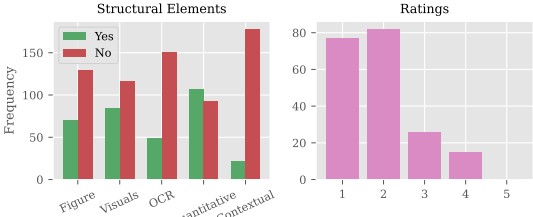

| Variant | BLEU-4↑ | ROUGE-L↑ | STS↑ | Length |
|---------|---------|----------|------|--------|
| Captions | 0.003 | 0.098 | 0.355 | 34.0 |
| Qwen2.5-VL-7B | 0.068 | 0.276 | 0.744 | 126.3 |
| Qwen2.5-VL-32B | 0.047 | 0.242 | 0.719 | 177.8 |
| InternVL3-8B | 0.045 | 0.235 | 0.716 | 159.8 |
| InternVL3-38B | 0.057 | 0.264 | 0.743 | 141.6 |
| GPT-4o-mini | 0.073 | 0.281 | 0.761 | 140.9 |
| GPT-4o | 0.089 | 0.317 | 0.777 | 123.5 |
| Human | 0.094 | 0.318 | 0.815 | 105.3 |

Figure 1: Left: human evaluation of caption quality by structural elements and usefulness ratings. Right: BLEU-4, ROUGE-L, STS, and average length for raw captions, VLM-generated descriptions, and human-written descriptions (using other human descriptions as references).

## 2 RELATED WORK

**Text-Guided Graphics Program Generation for Scientific Figures**    Generating vector graphics such as SVG or TikZ is essential in scientific publishing due to their fidelity and interpretability. Early approaches relied on handcrafted heuristics or neural sequence models to approximate images with path primitives (Lopes et al., 2019; Carlier et al., 2020), but these struggled with complex scientific figures. More recently, LLM-based methods have emerged: AutomaTikZ (Belouadi et al., 2024a) finetunes on caption–TikZ pairs from arXiv and TeX SE, while StarVector (Rodriguez et al., 2024) focuses on SVG generation with a dedicated benchmark. Yet for TikZ, dataset sparsity remains a bottleneck. TikZero (Belouadi et al., 2025) partially addresses this by combining an inverse-graphics model (Belouadi et al., 2024b) with a modality-bridging adapter (Hu et al., 2023), distilling supervision from text–image pairs. However, TikZero still depends on noisy captions and cannot finetune its text decoder without paired graphics programs, limiting performance. In contrast, we construct a dataset over four times larger and of higher quality, pairing TikZ programs with VLM-generated descriptions, enabling small LLMs to be effectively finetuned for Text-to-TikZ.

**Post-training with Reinforcement Learning**    Advances in RL such as Group Relative Policy Optimization (GRPO) (Shao et al., 2024) allow to more efficiently align LLMs either with human preferences (Ouyang et al., 2022) or verifiable tasks (Lambert et al., 2025). For example, RLEF (Gehring et al., 2025) iteratively leverages execution feedback for code synthesis,  Yoshihara et al. (2025) enhance LLM reasoning on math benchmarks, and VisionR1 (Huang et al., 2025) extends reasoning capabilities to the multimodal domain. Closest to our setting, RLRF (Rodriguez et al., 2025) optimizes SVG code generation via composite rewards assessing code efficiency, semantic alignment, and visual fidelity. Our work differs in two ways: we focus on TikZ generation for scientific figures, and we introduce a domain-specific reward model, trained through inverse-graphics (Image–TikZ), which better captures semantics than general-purpose metrics such as CLIPScore (Hessel et al., 2021) or DreamSIM (Fu et al., 2023).

## 3 CAPTION QUALITY ANALYSIS

Accurate Text-to-TikZ generation requires captions that specify objects, attributes, and spatial relations (Zhang et al., 2025). To assess whether existing captions meet this need, we analyzed 200 samples from DaTikZ-V3 with three annotators (Figure 1, left). The annotators checked captions for missing structural elements (e. g. figure type, components, and labels) and judged usefulness on a 1–5 Likert scale. Two findings emerged: (i) key details such as figure types, components, and labels are often missing, and (ii) most captions received low usefulness scores (1–2). This indicates that raw captions are insufficient for faithfully reconstructing scientific figures.

To quantify this further, a human annotator wrote reference descriptions for all 200 figures. We then compared these human-written descriptions against both the original captions and VLM-generated descriptions using BLEU-4 (Papineni et al., 2002), ROUGE-L (Lin, 2004), and Semantic Textual Similarity (STS) (Reimers & Gurevych, 2019) (Figure 1, right). Across multiple VLMs (Qwen2.5-VL 7B/32B (Bai et al., 2025), InternVL3 8B/38B (Zhu et al., 2025), GPT-4o and GPT-4o-mini (OpenAI et al., 2024)), results show that VLMs produce richer and more faithful descriptions than raw

captions. For example, GPT-4o reaches 0.089 BLEU-4 compared to just 0.003 for captions, and comes close to human-human agreement (0.094 BLEU-4). VLM outputs are also substantially longer (120–170 vs. 34 characters), indicating that they capture additional detail necessary for figure reconstruction. These results motivate our use of VLM-generated descriptions in DaTikZ-V4. For additional information, we refer to the Appendix A.1.

## 4 DATASET

Building on DaTikZ-V3, we introduce DaTikZ-V4, a significantly expanded and refined dataset designed to support the training and evaluation of Text-to-TikZ models. The development of DaTikZ-V4 addresses the growing need for both larger and higher-quality datasets, which are critical for surpassing not only proprietary state-of-the-art models like GPT-5 but also increasingly more capable open-source LLMs such as Qwen3.

**Data Sourcing** To enhance dataset scale, we first identify GitHub as a valuable large-scale source of high-quality graphics programs. With over one billion repositories, GitHub hosts a wealth of educational resources, tutorials, theses, books, and personal projects, many of which contain TikZ code. From this, we clone approximately 5,500 repositories containing `.tex` or `.pgf` files with TikZ content, resulting in over 400,000 unique TikZ samples. This GitHub-only subset is nearly as large as the entirety of DaTikZ-

Table 2: Unique TikZ graphics across all DaTikZ versions.

| Source | DaTikZ | V2 | V3 | V4 |
|---|---|---|---|---|
| arXiv | 85,656 | 326,450 | 407,851 | 1,471,083 |
| GitHub | 0 | 0 | 0 | 413,178 |
| TeX SE | 29,238 | 30,609 | 42,654 | 97,909 |
| Synthetic | 1,957 | 1,958 | 2,256 | 13,514 |
| Curated | 981 | 1,566 | 3,646 | 5,196 |
| **Total** | 117,832 | 360,583 | 456,407 | 2,000,880 |

V3. To further expand coverage, we also extend sourcing from arXiv by including data post-2021 to mid 2025. The increasing amounts of arXiv submissions each year allows us to source 1M additional samples, resulting in over 2M TikZ graphics in total. Of these, 35.55% originate from sources under permissive Creative Commons licenses (e.g., CC-BY, CC-BY-SA, CC0) and can be redistributed. 40.03% originate from sources under Nonexclusive-Distribution licenses, and the remaining 24.43% contain no explicit license information. A comparison of DaTikZ-V4 to previous releases is seen in Table 2.

**Filtering** Beyond traditional `tikzpicture` environments, we now extract from other environments such as `tikz-cd` (common in mathematical diagrams) and `circuitikz` (used in electronics). Since individual figures often contain multiple subfigures, we recursively split and extract all subfigure content. Furthermore, we enforce a standardized TikZ code by wrapping the code inside the `\documentclass[tikz]{standalone}` environment. Additionally, we implement a dynamic package detection approach by using regular expressions to include necessary LaTeX packages (e.g., recognizing `circuitikz` from context such as `resistor`). We also remove any code that depends on external files (e.g., `\input{...}`, `\includegraphics{...}`), as well as all inline comments, to improve compilation rates and reduce noise. Lastly, we apply exact deduplication and dismiss all samples where the number of characters is both smaller than 100 and larger than 4000.

**LLM Debugging** Due to the low compilation success rate, especially from arXiv (success rate 31.3%), we introduce an LLM-based debugging pipeline. Given a code snippet and its compiler error, an LLM is instructed to fix the TikZ code. Using Qwen-32B across our corpus of 1.3M uncompilable TikZ samples, we successfully repair 600K instances in the first pass. This approach substantially boosts the proportion of usable TikZ programs at scale.

**VLM-based Image Description** As shown in Section 3, raw captions are often unhelpful for figure reproduction, potentially leading to severe hallucinations. To mitigate this, we employ VLMs to generate precise descriptions of TikZ figures. Using Qwen2.5-VL-7B-Instruct, we annotate around 1.3M compilable samples, producing the first large-scale dataset of TikZ paired with semantically rich textual descriptions, providing stronger supervision for downstream model training. An overview of our dataset construction is illustrated in Figure 2. For ablations and further details about prompts and frameworks, we refer to A.2.

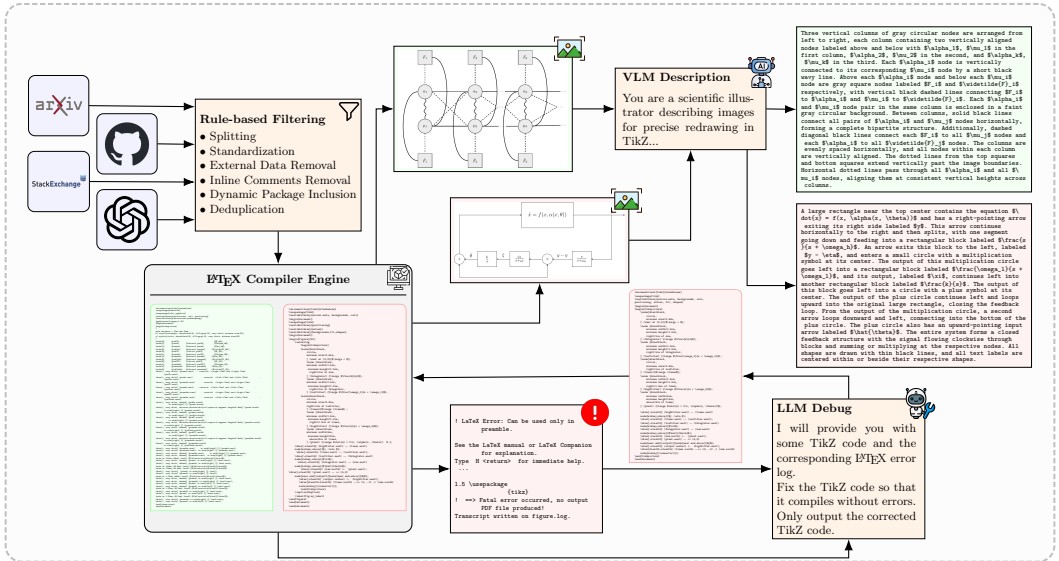

Figure 2: Overview of the data preprocessing workflow. We start by sourcing TikZ graphics programs primarily from arXiv, GitHub, TeX SE, as well as synthetic data. Next, rule-based filtering techniques are applied, and the TikZ code is rendered. Uncompilable code undergoes an iterative debugging process using LLMs alongside the error messages to attempt error correction. Finally, all compilable code images are described using VLMs.

## 5 METHOD

We train Text-to-TikZ models in two stages: SFT to ground models in TikZ syntax and task-specific token distributions, followed by RL for incorporating feedback from rendered images to enforce enhanced visual alignment (Rodriguez et al., 2025). Similar two-stage paradigms have also proven effective in related domains such as code generation and mathematical reasoning, where surface-level syntax is complemented by execution-level accuracy (Le et al., 2022; Gehring et al., 2025).

**Stage 1: Supervised Finetuning**    Given a figure description $x_{\text{desc}}$ and ground-truth TikZ sequence $x_{\text{tikz}} = (x_1, \ldots, x_T)$, we minimize the standard autoregressive negative log-likelihood:

$$\mathcal{L}_{\text{SFT}}(\theta) = \mathbb{E}_{(x_{\text{desc}}, x_{\text{tikz}}) \sim \mathcal{D}} \left[ - \sum_{t=1}^{T} \log p_\theta(x_t \mid x_{<t}, x_{\text{desc}}) \right] \tag{1}$$

This ensures syntactic validity and prompt alignment. At the same time, the model remains unaware of the rendered semantics of the figure, which leads to common errors such as loops, irrelevant content, or incorrect spatial relations.

**Stage 2: Reinforcement Learning**    To address this, we reinterpret the SFT model $p_{\theta_{\text{SFT}}}$ as a stochastic policy and apply reinforcement learning with GRPO. For each description, $G$ rollouts $\{o_1, \ldots, o_G\} \sim p_{\theta_{\text{old}}}(\cdot \mid x_{\text{desc}})$ are sampled, each of which is assigned a scalar reward $\{r_1, \ldots, r_G\}$ scored by a reward model, and updated with group-centered advantages $A_i = \frac{r_i - \text{mean}(\{r_j\})}{\text{std}(\{r_j\})}$. The GRPO objective we maximize is:

$$\mathcal{J}_{\text{GRPO}}(\theta) = \mathbb{E}_{x_{\text{desc}} \sim \mathcal{D}} \left[ \frac{1}{LG} \sum_{i=1}^{G} \sum_{t=1}^{|o_i|} \min\left( \frac{p_\theta(o_i \mid x_{\text{desc}})}{p_{\theta_{\text{old}}}(o_i \mid x_{\text{desc}})} A_i, \right. \right.$$

$$\left. \left. \text{clip}\left( \frac{p_\theta(o_i \mid x_{\text{desc}})}{p_{\theta_{\text{old}}}(o_i \mid x_{\text{desc}})}, 1 - \epsilon_{\text{low}}, 1 + \epsilon_{\text{high}} \right) A_i \right) - \beta \, D_{\text{KL}}\big(p_\theta \,\|\, p_{\theta_{\text{SFT}}}\big) \right]$$

where $\beta$ regulates the KL penalty. We implement the "Dr.GRPO" variant (Liu et al., 2025), which replaces the response-level normalization by a token-level normalization with a constant divisor

(the maximum completion length $L$). This removes the response length bias in TikZ sequences, where longer responses are under-penalized. Furthermore, we apply the "Clip-Higher" strategy from DAPO (Yu et al., 2025), which decouples the clipping threshold $\epsilon$ into $\epsilon_{\text{low}}$ and $\epsilon_{\text{high}}$. This allows more headroom for increasing the probability of low-probability exploration tokens (by raising $\epsilon_{\text{high}}$), while still preventing collapse of high-probability exploitation tokens (by keeping $\epsilon_{\text{low}}$ smaller). As in DAPO, we set $\epsilon_{low} = 0.2$ and $\epsilon_{high} = 0.28$. Additionally, we remove scaling the advantages by the standard deviation of the group rewards to not introduce a bias towards more or less difficult prompts, and mask all samples whose completion was cut by the length cap as we find that it increases training stability. Finally, we disable the KL coefficient ($\beta = 0$) and sample with `temperature=1.0` and `top_p=0.9`.

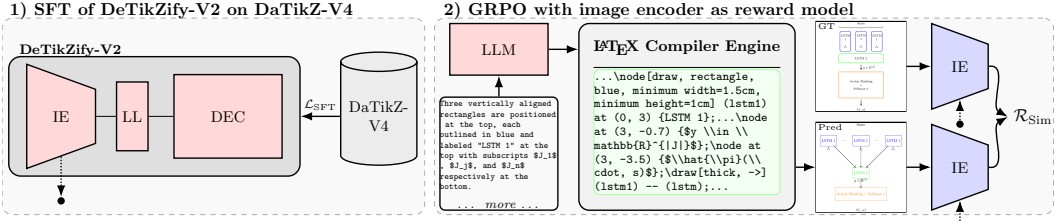

Figure 3: Overview of our post-SFT optimization steps. We first fully finetune DeTikZify-V2 consisting of an image encoder (IE), linear layer (LL) and LLM decoder (DEC) on our larger DaTikZ-V4 where we then use its enhanced IE to further finetune our LLMs based on the semantic similarity of the embeddings between ground truth and rendered image in an online RL setting using GRPO. The IE is kept frozen during RL optimization to mitigate reward hacking.

**Rewards** Designing reward signals for Text-to-TikZ is challenging: they must capture faithfulness, scientific style, attributes, and spatial relations. Recent work has shown that metrics such as CLIPScore or DreamSim correlate poorly with human judgments as they fail to represent nuances in scientific figures (Belouadi et al., 2025) and are prone to reward hacking (e.g., embedding text into figures) (Rodriguez et al., 2025).

To the best of our knowledge, we propose the first domain-specific reward model for Text-to-TikZ. It builds on the image encoder of DeTikZify-V2 (Belouadi et al., 2024b), originally trained on DaTikZ-V3 for inverse graphics (image → TikZ). DeTikZify consists of an image encoder followed by a linear layer and an LLM decoder. By keeping the image encoder unfrozen during training, it incidentally learns to generate good low-dimensional representations of scientific figures in order to accurately reproduce the figure, allowing us to utilize it to measure semantic similarity between the embeddings of two scientific figures more accurately. With DaTikZ-V4 providing a much larger dataset, we retrain DeTikZify-V2 end-to-end, yielding a stronger encoder that produces richer, more generalizable embeddings of scientific diagrams. Subsequently, we use the retrained image encoder as our reward model in an online RL environment with GRPO. Both steps are illustrated in Figure 3. Training details are provided in A.3.

For reward computation, pooled cosine similarity is not available since DeTikZify-V2 outputs patch-level embeddings. We therefore adopt an Earth Mover's Distance (EMD) (Rubner et al., 1998; Kusner et al., 2015) formulation, inspired by test-time scaling approaches in TikZero (Belouadi et al., 2025). Given patch embeddings $\mathbf{x} = \{x_i\}_{i=1}^{|\mathbf{x}|}$ and $\mathbf{y} = \{y_j\}_{j=1}^{|\mathbf{y}|}$ from ground truth and predicted images, with distance matrix $D_{i,j} = 1 - \cos(x_i, y_j)$, the similarity reward is defined as

$$\mathcal{R}_{\text{Sim}}(\mathbf{x}, \mathbf{y}) = 1 - \frac{\sum_{i=1}^{|\mathbf{x}|} \sum_{j=1}^{|\mathbf{y}|} F_{i,j} D_{i,j}}{\sum_{i=1}^{|\mathbf{x}|} \sum_{j=1}^{|\mathbf{y}|} F_{i,j}}, \tag{2}$$

where $F \in \mathbb{R}_{\geq 0}^{|\mathbf{x}| \times |\mathbf{y}|}$ is the optimal flow matrix that minimizes the transport cost, subject to $\sum_i F_{i,j} = 1/|\mathbf{y}|$ and $\sum_j F_{i,j} = 1/|\mathbf{x}|$. This formulation yields a scalar reward in $[0, 1]$ capturing semantic alignment. Finally, we add a format reward to ensure that the TikZ code starts and ends with valid document environments (i.e., `\documentclass[tikz]{standalone}`, followed by `\begin{document}`, and ending with `\end{document}`). Non-conforming outputs receive a reward of zero.

## 6 EXPERIMENTS

**Experimental Setup** For evaluation, we construct a contamination-free test set of 1,047 samples from DaTikZ-V4. To prevent overlap with training data, we (i) restrict to post–May 2025 samples, (ii) enforce per-source uniqueness (e.g., one figure per arXiv paper or GitHub repo, removing the rest from training), (iii) filter with n-gram matching (OpenAI, 2023), and (iv) manual inspection to discard trivial or corrupted figures. To avoid model bias, all test descriptions are generated by GPT-4o. For RL-tuning, we create DaTikZ-V4-RL, a 160K-sample subset obtained by repairing uncompilable figures via a second LLM debugging step and re-describing them with Qwen2.5-VL-7B. This provides additional high-quality pairs beyond the training split.

**Models** We benchmark nine LLMs: (i) proprietary GPT-5[1] and GPT-4o (OpenAI et al., 2024), (ii) open-source Qwen3 (32B, 8B), Qwen3-Coder-30B-A3B (Yang et al., 2025), Qwen2.5 (14B, 3B) (Qwen et al., 2025), TikZero-Plus-10B (Belouadi et al., 2025), and Llama3.1-8B (Grattafiori et al., 2024), and (iii) our fine-tuned Qwen2.5-3B and Qwen3-8B models. We refer to our trained models as TikZilla, with the following variants: TikZilla-3B and TikZilla-8B (SFT only), and TikZilla-3B-RL and TikZilla-8B-RL (two-stage training). In addition, we also test RL-only training.

**Evaluation Metrics** We evaluate along four axes: (i) CLIPScore (CLIP) (Hessel et al., 2021) for text–image alignment, (ii) DreamSIM (DSim) (Fu et al., 2023) for perceptual fidelity, (iii) TeX Edit Distance (TED) (Kusner et al., 2015) for code similarity, and (iv) Compilation Rate (CR) for executability. We also report average tokens (AT) for efficiency. To avoid reward–metric coupling in our RL ablations, we additionally report DINOScore (DINO) (Caron et al., 2021) and LPIPS (Zhang et al., 2018), which are independent of our domain-specific reward model. An aggregate score (AVG) is computed as the mean of CLIP/DINO, DSim/LPIPS, and 1-TED (depending on the evaluation setting). Additional details are reported in A.4.

## 7 RESULTS

**Main Results** Table 3 reports results on automatic metrics. Our SFT+RL-tuned Qwen models achieve the best AVG performance, with TikZilla-3B-RL reaching 0.385 and TikZilla-8B-RL 0.384. Both surpass GPT-5 (0.365), despite it being recently released as one of the strongest reasoning LLMs, evaluated with no output length restrictions. Compared to the recently released TikZero-Plus-10B, TikZilla-3B-RL improves by +0.085 on CLIP and +0.334 on DSim, while achieving a 37% higher compilation rate and requiring 261 fewer tokens on average. Similar improvements hold for TikZilla-8B-RL. These results highlight the effectiveness of our two-stage training process, combining high-quality data with a domain-specific reward model. For qualitative examples with TikZ code, we refer to the Appendix (Figure 15, 16, 17, and 18).

MODEL SIZE AND TRAINING REGIME Interestingly, the smaller Qwen2.5-3B not only closes the gap with Qwen3-8B but even slightly outperforms it once trained with SFT+RL. However, its low baseline (0.202) indicates that it strongly relies on SFT before RL, whereas Qwen3-8B benefits from RL directly ($0.251 \rightarrow 0.357$). This suggests that SFT primarily provides syntax grounding for smaller models, while larger models already encode some TikZ knowledge that RL can amplify.

IMPLICIT EFFICIENCY EFFECTS RL consistently improves compilation rates to 95–98% and reduces token length, indicating more efficient code generation. Unlike prior SVG studies (Rodriguez et al., 2025), which required explicit code efficiency rewards, we observe a natural reduction in sequence length. We hypothesize this stems from our semantic reward model penalizing hallucinated or redundant elements, indirectly encouraging conciseness. A deeper comparison with explicit efficiency rewards is left for future work.

**Human Evaluation** We conduct a human evaluation with 9 expert annotators (6 PhD, 2 postdoc, 1 faculty member). Each annotator rated 30 randomized figures/descriptions across 4–5 models, using a 1–5 Likert scale (1 = uncompilable, 5 = publication-ready). Two criteria were considered:

---

[1]https://openai.com/de-DE/index/gpt-5-system-card/

Table 3: Results of all models on the evaluation subset of DaTikZ-V4. Both of our models trained with SFT and RL perform best, while GPT-5 and Qwen3-32B are the best proprietary and open-source LLMs respectively. **Bold** denotes best-performing while underline is second-best.

| LLM | CLIP↑ | DSim↑ | TED↓ | AVG↑ | CR↑ | AT |
|---|---|---|---|---|---|---|
| GPT-5 | 0.181 | 0.679 | 0.765 | 0.365 | 88% | 480 |
| GPT-4o | 0.147 | 0.580 | 0.767 | 0.320 | 78% | 404 |
| Qwen3-32B | 0.149 | 0.583 | 0.765 | 0.322 | 79% | 416 |
| Qwen3-Coder-30B-A3B | 0.140 | 0.566 | 0.778 | 0.309 | 77% | 472 |
| Qwen2.5-14B | 0.132 | 0.511 | 0.765 | 0.293 | 71% | 376 |
| TikZero-Plus-10B | 0.104 | 0.397 | 0.807 | 0.231 | 61% | 742 |
| Llama3.1-8B | 0.088 | 0.339 | 0.786 | 0.214 | 50% | 529 |
| Qwen2.5-3B | 0.081 | 0.315 | 0.789 | 0.202 | 52% | 387 |
| Qwen2.5-3B (+RL) | 0.098 | 0.505 | 0.795 | 0.269 | **98%** | 234 |
| TikZilla-3B | 0.161 | 0.613 | 0.802 | 0.324 | 89% | 672 |
| TikZilla-3B-RL | **0.189** | **0.731** | 0.766 | **0.385** | **98%** | 481 |
| Qwen3-8B | 0.106 | 0.421 | 0.775 | 0.251 | 63% | 412 |
| Qwen3-8B (+RL) | 0.169 | 0.669 | 0.768 | 0.357 | **98%** | 393 |
| TikZilla-8B | 0.158 | 0.602 | 0.793 | 0.322 | 86% | 729 |
| TikZilla-8B-RL | 0.185 | 0.727 | **0.761** | 0.384 | 95% | 459 |

(i) textual alignment (does the output follow the provided description?) and (ii) image alignment (does the output match the original ground-truth figure?). Annotator agreement was strong (Cohen's $\kappa = 0.814$ for text, 0.794 for image). Full details are provided in A.5.

RESULTS   Figure 4 shows that GPT-5 achieved the highest textual score (4.18) and tied with our TikZilla-8B-RL on image evaluation (3.48 vs. 3.46). TikZilla-3B-RL also performed competitively (3.40 text, 3.30 image). Reinforcement learning substantially boosted both Qwen models (+0.75 and +0.67 points), while base models lagged 1.5–2 points behind. Interestingly, most models (especially GPT-5) scored higher on the textual evaluation than on the image evaluation. We hypothesize two possible explanations: (i) if VLM-generated captions omit or misrepresent visual details, models may score highly on textual alignment (satisfying the description) but lower on image alignment (failing to match the true figure). (ii) Human annotators may apply stricter criteria when comparing against ground-truth images than when comparing against text. Disentangling these two factors remains an open question, which we leave for future work.

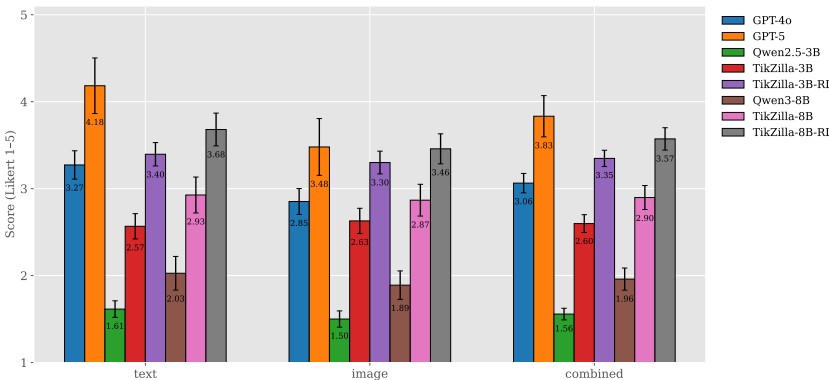

Figure 4: Average Likert-scale ratings (1–5, higher is better) with 95% confidence intervals for eight LLMs, evaluated under two settings: (i) alignment with textual descriptions and (ii) alignment with ground-truth images. Combined scores are shown as the average of both settings.

CORRELATION WITH METRICS   Finally, we compute correlations between automatic metrics and human scores using Spearman's $\rho$. CLIP ($\rho_{CLIP} = 0.260$) and TED ($\rho_{1-TED} = 0.307$) show

Table 4: Ablations on input data quality and debugging. VLM-based descriptions consistently outperform captions, while mixing or oversampling captions brings no gains. Our LLM-based debugging step yield improvements.

| LLM | CLIP↑ | DSim↑ | TED↓ | AVG↑ | CR↑ | AT |
|---|---|---|---|---|---|---|
| GPT-4o$_{cap.}$ | 0.105 | 0.469 | 0.763 | 0.270 | 80% | 337 |
| GPT-4o$_{desc.}$ | 0.143 | 0.568 | 0.767 | 0.315 | 76% | 416 |
| Qwen2.5-3B (+SFT$_{cap.}$) | 0.134 | 0.511 | 0.809 | 0.279 | 79% | 768 |
| Qwen2.5-3B (+SFT$_{desc.}$) | 0.141 | 0.530 | 0.805 | 0.289 | 85% | 651 |
| Qwen2.5-3B (+SFT$_{desc. \lor cap.}$) | 0.154 | 0.589 | 0.804 | 0.313 | 85% | 735 |
| Qwen2.5-3B (+SFT$_{desc. + cap.}$) | 0.157 | 0.599 | 0.799 | 0.319 | 89% | 787 |
| Qwen2.5-3B (+SFT$_{no debug}$) | 0.138 | 0.534 | 0.809 | 0.288 | 79% | 762 |
| TikZilla-3B | 0.161 | 0.613 | 0.802 | 0.324 | 89% | 672 |

Table 5: Ablation of our domain-specific reward models compared to CLIP$_{Img}$ and DreamSIM. The DaTikZ-V4 trained encoder achieves the strongest AVG performance.

| LLM | DINO↑ | LPIPS↑ | TED↓ | AVG↑ | CR↑ | AT |
|---|---|---|---|---|---|---|
| Qwen2.5-3B (+SFT+RL$_{CLIP_{Img.}}$) | 0.751 | 0.418 | 0.779 | 0.463 | 97% | 537 |
| Qwen2.5-3B (+SFT+RL$_{DSim}$) | 0.759 | 0.439 | 0.777 | 0.474 | 99% | 494 |
| Qwen2.5-3B (+SFT+RL)$_{\mathcal{R}_{Sim}(DaTikZ-V3)}$ | 0.789 | 0.440 | 0.768 | 0.487 | 97% | 496 |
| TikZilla-3B-RL | 0.809 | 0.451 | 0.766 | 0.498 | 98% | 481 |

weak, DSim moderate ($\rho_{DSim} = 0.586$), and our reward model strong ($\rho_{\mathcal{R}_{Sim}} = 0.714$) correlation. This validates our design of a domain-specific reward model aligned with human judgment.

**Ablations** We perform a series of ablations to isolate the contribution of each component in our pipeline. These include: (i) the impact of input quality and LLM-based debugging (Table 4), (ii) the effect of different reward models (Table 5), and (iii) the influence of dataset scale (Figure 5). We additionally evaluate TikZilla in an out-of-distribution (OOD) setting using the SPIQA (Pramanick et al., 2024) benchmark (Table 6).

CAPTIONS VS. DESCRIPTIONS VLM-generated descriptions consistently outperform raw captions. At inference, GPT-4o achieves 0.315 AVG with descriptions versus 0.270 with captions, confirming our earlier analysis that captions are often unhelpful for figure reproduction. Examples are shown in Figure 14 in the Appendix. For training, Qwen2.5-3B also benefits from descriptions (0.289 vs. 0.279), though the gap is smaller, likely due to the limited caption subset (468k samples). Mixing captions/descriptions (desc. ∨ cap.) and oversampling descriptions with captions (desc. + cap.) degrade performance, suggesting that low-quality captions dilute training even when more data is added.

LLM-BASED DEBUGGING Models trained only on first-try compilable code perform considerably worse than those trained on the full dataset (0.288 vs. 0.324), highlighting the necessity of our LLM-based debugging pipeline to increase the size of our usable TikZ corpus.

REWARD MODEL TRAINING We compare our domain-specific reward $\mathcal{R}_{Sim}$ against CLIPScore (image–image) and DreamSIM (Table 5). We find that all reward functions improve over the SFT baseline, but $\mathcal{R}_{Sim}$ achieves the strongest AVG performance. DreamSIM performs slightly better than CLIPScore. Retraining DeTikZify-V2 on DaTikZ-V4 yields a stronger reward model (0.389 vs. 0.375). Correlations with human judgments also improve ($\rho_{\mathcal{R}_{Sim}} = 0.714$ vs. 0.698), confirming that larger-scale scientific data produces more reliable image encoders for semantic evaluation.

DATASET SIZES To understand how performance scales with data, we supervised fine-tune Qwen2.5-3B on subsets of DaTikZ-V4 at 75%, 50%, 25%, 12.5%, and 6.25% of the full dataset (Figure 5). Performance increases sharply at small data scales (from 0–25%), after which improvements become more gradual from 25% up to the full dataset, suggesting that further data scaling (e.g., synthetic data) remains a promising direction for improving text-to-TikZ generation.

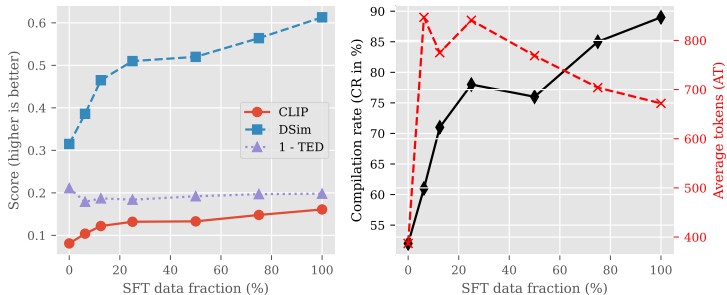

Figure 5: SFT on Qwen2.5-3B with different dataset scales (75%, 50%, 25%, 12.5%, and 6.25%).

Table 6: Model performance on the SPIQA benchmark.

| LLM | CLIP↑ | DSim↑ | CR↑ | AT |
|---|---|---|---|---|
| GPT-5 | 0.115 | 0.432 | 60% | 1239 |
| GPT-4o | 0.098 | 0.326 | 48% | 748 |
| Qwen3-Coder-30B-A3B | 0.102 | 0.349 | 58% | 1000 |
| Qwen2.5-3B | 0.038 | 0.117 | 24% | 772 |
| TikZilla-3B | 0.114 | 0.374 | 64% | 1170 |
| TikZilla-3B-RL | **0.193** | **0.637** | **97%** | 765 |
| Qwen3-8B | 0.070 | 0.228 | 37% | 781 |
| TikZilla-8B | 0.131 | 0.428 | 70% | 1184 |
| TikZilla-8B-RL | 0.174 | 0.584 | 90% | 809 |

OOD DATA    To assess TikZilla's robustness under distribution shift, we evaluate it on the SPIQA dataset. SPIQA figures are typically not generated in TikZ but originate from tools such as mat-plotlib, ggplot2, and MATLAB, often containing multi-panel layouts, overlays, and varied diagram-matic structures. This makes SPIQA a meaningful OOD benchmark from a structural-complexity perspective. For evaluation, we use all samples only including figures from the test-A and test-B splits and generate textual descriptions with GPT-4o, yielding 397 test cases. Since ground-truth TikZ code is unavailable, we omit TED in this evaluation. Relative to the DaTikZ-V4 test split (Table 3), SPIQA exhibits substantially longer sequences, lower compilation rates, and overall lower performance, as expected due to its non-TikZ origin and higher visual complexity. Notably, both TikZilla-8B-RL and especially TikZilla-3B RL outperform GPT-5 on this OOD benchmark.

## 8    CONCLUSION, LIMITATIONS, AND FUTURE WORK

We presented DaTikZ-V4, a large-scale, high-quality dataset for Text-to-TikZ, and a two-stage train-ing framework combining SFT with RL. Our key contributions are a richer dataset sourced from arXiv and GitHub with LLM-based debugging to improve compilability, VLM-generated descrip-tions that overcome the low quality of raw captions, and a domain-specific reward model derived from an inverse-graphics image encoder, which correlates strongly with human judgments of figure quality. Building on these components, we introduced TikZilla, a family of small Qwen-based mod-els that achieve near-perfect compilation rates and even surpass much larger commercial systems such as GPT-4o across automatic and human evaluation. Beyond technical performance, TikZilla demonstrates the feasibility of building reproducible and efficient text-to-image generation systems with small-scale open models, reducing reliance on costly proprietary solutions.

A key limitation is that our figure descriptions are generated automatically by VLMs, which may introduce omissions or hallucinations. This can bias training and, in rare cases, reward optimization may reinforce errors when descriptions diverge from figures. More reliable annotation methods and fine-grained reward functions are therefore crucial directions for future work. Beyond addressing these issues, future work should focus on designing automatic metrics with stronger alignment to human perception, and extending our approach to other structured generation tasks (e.g., LaTeX tables, CAD, or flowcharts), where programmatic fidelity is critical.

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

# A APPENDIX

## A.1 CAPTION QUALITY ANALYSIS

Our caption quality analysis involved three annotators: one bachelor's student, one PhD student, and one faculty member (all male). From our subset of DaTikZ-V3, 74% of samples originate from arXiv and 26% from TeX SE. One annotator completed the evaluation sheet in Figure 6, based on the taxonomy in Table 7. This annotator also manually described all 200 scientific figures, which we subsequently used as reference descriptions to compute BLEU-4, ROUGE-L, and STS with the `all-mpnet-base-v2` sentence encoder between human descriptions and VLM-generated descriptions.

The other two annotators each described 30 figures to measure agreement, yielding unweighted $\kappa = 0.35$ and weighted $\kappa = 0.63$. The structural elements for scientific figure captions were adapted from best practices in academic writing and prior research taxonomies (Tang et al., 2023; Hsu et al., 2024).

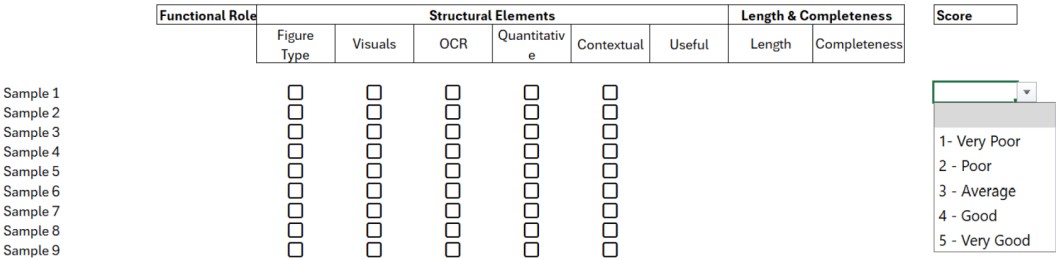

Figure 6: Screenshot of our evaluation form for the first nine scientific figures.

Table 7: Caption analysis taxonomy for structural elements and usefulness scores.

| Structural Elements | **Figure type:** names the high-level type (e.g., graph, tree, workflow). |
| --- | --- |
| | **Visual details:** mentions colors, shapes, axes, layout/spatial relations. |
| | **OCR:** includes textual elements visible in the figure (axis labels, annotations, math), aiding correct labeling. |
| | **Contextual reference:** points outside the figure (e.g., "see Sec. 3"). Useful but reduces standalone utility. |
| | **Quantitative content:** numbers, formulas, code. Adds technical substance (often paired with OCR). |
| Usefulness Scores | **Very Poor:** not meaningfully descriptive. May be only a label or irrelevant text. |
| | **Poor:** somewhat relevant but vague/incomplete. Mentions topic/elements without adequate clarity or context. |
| | **Average:** describes the main content but lacks depth/specifics. States what it is without highlighting key details. |
| | **Good:** clear, specific, and near-complete. Covers important visual/quantitative details and structure. |
| | **Very Good:** precise, insightful, and largely self-contained. Explains key elements so the figure is almost unnecessary. |

## A.2 DATASET

To create synthetic data, we follow a strategy similar to ScImage (Zhang et al., 2025). We first generate 2,000 templates with varied terms, each used to produce 10 queries that generate TikZ code. All steps are performed using GPT-4o with minimal human intervention.

**LLM Debugging** For LLM-based debugging, we use the prompt in Figure 7. We first tested this on a subset of 753 samples spanning all sources, manually evaluating the percentage of compilable, non-empty, and non-corrupted outputs. As shown in Table 8, Qwen3-32B (non-thinking) was the best-performing model, recovering 49.40% of errors in a single pass and 59.04% after three repair rounds. Smaller Qwen variants and Qwen2.5-7B-Instruct (Qwen et al., 2025) performed considerably worse. We therefore applied Qwen3-32B for large-scale debugging, which took 14 days on 4 × A100 40GB GPUs using the vLLM framework (Kwon et al., 2023). Examples of the debugging process are shown in Figure 8 and 9.

Table 8: Accuracy of different LLMs in debugging TikZ code from error logs over three refinement iterations. Bold indicates the best-performing model.

| LLM | Iteration 1 | Iteration 2 | Iteration 3 |
|---|---|---|---|
| Qwen2.5-7B | 17.49% | 28.03% | 34.08% |
| Qwen3-4B | 14.17% | 18.56% | 21.98% |
| Qwen3-8B | 35.11% | 39.36% | 41.49% |
| Qwen3-32B | **49.40%** | **55.42%** | **59.04%** |
| GPT-4o-mini | 36.82% | 41.36% | 43.62% |
| GPT-4o | 48.10% | 53.73% | 58.89% |

**LLM Debug Prompt**

```
I will provide you with some TikZ code and the corresponding LaTeX error log.  Fix the
TikZ code so that it compiles without errors.  Only output the corrected TikZ code.\n
Original TikZ Code: {tikz_code}\n
Compilation Error Log: {log_message}
```

**VLM-based Image Description**    The prompt for image description is shown in Figure 10. We use few-shot in-context learning (Brown et al., 2020) with two high-STS human descriptions as exemplars. We run Qwen2.5-VL-7B-Instruct, which was the strongest open-source VLM in our evaluation, to describe all figures in DaTikZ-V4. Processing required 2 days on 4 × A100 40GB GPUs.

**VLM Description Prompt**

```
You are a scientific illustrator describing images for precise redrawing in TikZ.\n
Your task is to describe the image in precise, continuous prose without bullet points,
lists, or line breaks.\n
Start directly with the main object or scene.  Avoid introductory phrases like
'Certainly!', 'The image depicts...', 'Here is a precise description.'.\n
Use clear, active language focused on geometry, labels, colors, spatial relationships,
coordinates, and other visible properties.\n
Describe all visible elements such as shapes, lines, arrows, and labels, including
their relative or absolute positions, dimensions, and orientation.\n
Use consistent, minimal naming for objects (e.g., 'circle A', 'line L1') and specify
label positions relative to shapes precisely.\n
Only describe exact, concrete visual elements that enable precise image reconstruction
in TikZ.\n
Avoid vague, interpretive, or inferential language, and exclude summaries,
conclusions, or commentary about the image's meaning, function, or aesthetics.\n
Here are a few examples:\n
A thin black horizontal line centered in the middle, containing nine evenly spaced
black dots, and labeled x_2 at the left.  Each dot is connected by a thin black line in
an alternating pattern to either x_0 (placed at the top middle) or x_1 (placed at the
bottom middle).\n
A line chart has different instruction scales of 1/10, 1/4, 1/2, and 1 on the x-axis.
On the y-axis it shows BLEU scores between 20 and 50, with steps of 5.  The chart
contains three lines with Zh-En in blue, De-En in red, and Fr-En in brown.  All BLEU
scores are initially 20 at the lowest instruction scale.  As the instruction scale
increases, BLEU scores improve for all pairs.  De-En is the highest, closely followed
by Fr-En and then Zh-En far below.  The increase is largest from 1/10 to 1/4 and only
marginally above an instruction scale of 1/4.  The legend is placed inside the chart
at the top left.\n
Write a description in this exact style for the given image.
```

### A.3 METHOD

For finetuning DeTikZify-V2 (Belouadi et al., 2024b), which is a SigLIP (Zhai et al., 2023) vision encoder of `PaliGemma-3b-mix-448` (Beyer et al., 2024)), we use the training split of DaTikZ-V4 consisting of 1.3M Image–TikZ pairs. Inputs are 448×448-pixel images with a maximum output length of 2048 tokens. Training runs for two epochs with a learning rate of 5e-5, AdamW (Loshchilov & Hutter, 2019), cosine scheduler, and 3% linear warmup. The batch size is 128, trained on 4 × H200 140GB GPUs for 12 days.

**LLM-based TikZ Debugging**

**Original TikZ Code:**

```
\documentclass[tikz]{standalone}
\usepackage[utf8]{inputenc}
\usepackage{circuitikz}
\usepackage{float}
\usepackage{calc}
\begin{document}
\begin{circuitikz}[american, straight voltages]
  \draw (-1,0)
  to [american voltage source, v=$V_P$, invert, voltage shift=1] (-1,4)
  to [R, R=$R_p$, i^>=$i_p$] (2,4)
  to [R=$R_L$] (4,4)
  to [L, l_=$L$, v^<=$v_L$, i=$i_L$, voltage shift=1.5] (7,4)
  to [Tnigbt,bodydiode] (10,4)
  to [short] (12,4)
  to [american voltage source, v^<=$V_{out}$, voltage shift=1] (12,0)
  to [short] (-1,0)
  (2.0,4) to [R=$R_Ci$, i=$i_{Ci}$] (2.0,1.5)
  to [C, l_=$C_i$, v^<=$v_{Ci}$] (2.0,0)
  (7.2,4) to [Tnigbt,bodydiode, invert] (7.2,0)
  (10.0,4) to [R=$R_Co$, i=$i_{Co}$] (10.0,1.5)
  to [C, l_=$C_o$, v^<=$v_{Co}$] (10.0,0)
  (8.5,5) node[align=center]{$G_2$}
  (6.1,2) node[align=center]{$G_1$}
  (7.2,0) node[circ, scale=1.5]{$1$}
  (7.2,4) node[circ, scale=1.5]
  (2,0) node[circ, scale=1.5]
  (2,4) node[circ, color=red, scale=1.5]
  (10,4) node[circ, color=red, scale=1.5]
  (10,0) node[circ, color=red, scale=1.5]
  ;
\end{circuitikz}
\end{document}
```

**Compiler Error Log:**

```
! Package tikz Error: A node must have a (possibly empty) label text.
See the tikz package documentation for explanation.
Type H <return> for immediate help.
 ...
l.26 (2,0) node[circ, scale=1.5]
! ==> Fatal error occurred, no output PDF file produced!
Transcript written on figure.log.
```

**Corrected TikZ Code (Changed Parts):**

```
...
(7.2,4) node[circ, scale=1.5]{}
(2,0) node[circ, scale=1.5]{}
(2,4) node[circ, color=red, scale=1.5]{}
(10,4) node[circ, color=red, scale=1.5]{}
(10,0) node[circ, color=red, scale=1.5]{}
...
```

Figure 8: An example of the LLM debugging pipeline. The original TikZ code failed to compile. The compiler error log was passed to the LLM, which generated corrected TikZ code. The fixed code produces the valid figure shown above.

---

**LLM-based TikZ Debugging**

**Original TikZ Code:**

```
\documentclass[tikz]{standalone}
\usepackage{tikz}
\usetikzlibrary{automata,shapes.geometric}
\usepackage{array}
\begin{document}
\begin{figure}[h]
\begin{tabular}{*{2}{>{\centering\arraybackslash}b{\dimexpr0.5\textwidth-2\tabcolsep\relax}}}
\legend{Weighted, complete graph $K_H$}
\begin{tikzpicture}[state/.append style={minimum size=5mm}]
    \node [state] (0) at (-2, 3) [label=left:E] {};
    \node [state] (1) at ( 2, 3) [label=right:B]{};
    \node [state] (2) at (-1.25, 0.75)[label=left:D] {};
    \node [state] (3) at ( 1.25, 0.75) [label=right:C]{};
    \node [state] (4) at ( 0, 4.5) [label=above: A]{};
    \draw (0) to (4);
    \draw (4) to (1);
    \draw (1) to (3);
    \draw (0) to (2);
    \draw (3) to (2);
    \draw (2) to (4);
    \draw (4) to (3);
    \draw (0) to (1);
    \draw (0) to (3);
    \draw (2) to (1);
\end{tikzpicture} &
\renewcommand{\arraystretch}{1.3}
\legend{\textbf{THIS IS TABLE LEGEND}}
\begin{tabular}{c|ccccc}
 & A & B & C & D & E \\
\hline
A & -- & 4 & 7 & 6 & 12 \\
B & 4 & -- & 3 & 5 & 8 \\
C & 7 & 3 & -- & 2 & 5 \\
D & 6 & 5 & 2 & -- & 9 \\
E & 12 & 8 & 5 & 9 & --
\end{tabular}
\end{tabular}
\end{figure}
\end{document}
```

**Compiler Error Log:**

```
! LaTeX Error: Not allowed in LR mode.
See the LaTeX manual or LaTeX Companion for explanation.
Type H <return> for immediate help.
 ...
l.6 \begin{figure}[h]
! ==> Fatal error occurred, no output PDF file produced!
Transcript written on figure.log.
```

**Corrected TikZ Code (Changed Parts):**

```
...
\begin{document}
\begin{tabular}{*{2}{>{\centering\arraybackslash}b{\dimexpr0.5\textwidth-2\tabcolsep\relax}}}
\begin{tikzpicture}[state/.append style={minimum size=5mm}]
...
\renewcommand{\arraystretch}{1.3}
\begin{tabular}{c|ccccc}
...
\end{tabular}
\end{tabular}
\end{document}
```

Figure 9: An example of the LLM debugging pipeline. The original TikZ code failed to compile. The compiler error log was passed to the LLM, which generated corrected TikZ code. The fixed code produces the valid figure shown above.

## A.4 EXPERIMENTS

The prompt template for all models is shown in Figure 11. We also experimented with templates without the standalone environment but found that this reduced performance and compilation rates.

**Models** Except for GPT-5, decoding uses `temperature=1.0`, `top_p=0.9`, and max length 2048. For GPT-5, we set `reasoning=medium`, `verbosity=medium`, and evaluate a random subset of 100 samples due to cost. For TikZero, trained on caption–TikZ pairs, we only provide the figure description as prompt. For SFT, Qwen2.5-3B is finetuned on DaTikZ-V4 for two days with a learning rate of 1e-4, warmup ratio 3%, cosine scheduler, and batch size 128. Qwen3-8B is trained for four days with a reduced learning rate of 5e-5. For RL on DaTikZ-V4-RL, TikZilla-3B is trained with GRPO for 4,000 iterations (batch size 144, 8 rollouts) using learning rate 5e-6 and weight decay 1e-2. TikZilla-8B uses learning rate 2e-6. RL-only runs were also tested. Training took 5 days for TikZilla-3B and 10 days for TikZilla-8B, all on 4 × H200 140GB GPUs.

**Metrics** CLIPScore (**CLIP**) is computed with `siglip-so400m-patch14-384`. Dream-SIM (**DSim**) uses CLIP, DINO, and OpenCLIP (`ViT-B/16`). TeX Edit Distance (**TED**) uses `TexLexer`. Average tokens (**AT**) are measured with `o200k_base` tokenizer. DINOScore (**DINO**) is calculated using the cosine similarity of the patch embeddings produced by `dino-vits16`. Learned Perceptual Image Patch Similarity (**LPIPS**) uses the `alex` network.

```
Prompt Template

Generate a complete LaTeX document that contains a TikZ figure according to the
following requirements:
{figure_description}
Wrap your code using \documentclass[tikz]{standalone}, and include
\begin{document}...\end{document}.  Only output valid LaTeX code with no extra text.
```

## A.5 RESULTS

**Human Evaluation** We split 9 annotators (6 male, 3 female) into two groups. Group 1 (5 annotators) evaluated GPT-5, GPT-4o, Qwen2.5-3B, TikZilla-3B, and TikZilla-3B-RL. Group 2 (4 annotators) evaluated GPT-5, GPT-4o, Qwen3-8B, TikZilla-8B, and TikZilla-8B-RL. Each annotator received two Excel sheets (textual vs. image alignment), each with 30 randomized samples. We ensured at least five overlapping samples for inter-annotator agreement and five GPT-5 samples (scarcer due to cost). Annotation interfaces are shown in Figures 12 and 13. Likert scale definitions are shown below:

- **5 Excellent**: Figure fulfills all requirements. Few minor issues (e.g., slightly imperfect layout, one or two mislabeled/extra elements) are acceptable. Think about it as publication or almost publication ready where only small tweaks needs to be made.
- **4 Good**: Figure broadly fulfills the requirements and contains no major errors, but it is clearly not perfect. Typical cases include multiple minor flaws (e.g., clutter, small inaccuracies, awkward design) or one moderate issue.
- **3 Fair**: The figure has about one to two major issues (e.g., important elements missing, wrong trends in charts, ...) and/or some minor issues. It is still usable with corrections as parts of the figure are clearly correct.
- **2 Poor**: Several major issues and/or many minor ones. The figure no longer meaningfully reflects the description or GT image (e.g., severe overlaps, high amounts of hallucinated content, ...).
- **1 Failed**: Non-compilable code (already auto-assigned).

**Ablations** For inference, we ablate input quality by comparing GPT-4o on the evaluation subset where captions are available (GPT-4o$_{cap.}$) versus the same subset with VLM-generated descriptions instead (GPT-4o$_{desc.}$). For training, we finetune Qwen2.5-3B on different input variants: (i) Qwen2.5-3B (SFT$_{cap.}$), using only caption–TikZ pairs (468k samples), (ii) Qwen2.5-3B (SFT$_{desc.}$), using the same subset but replacing captions with VLM descriptions, (iii) Qwen2.5-3B

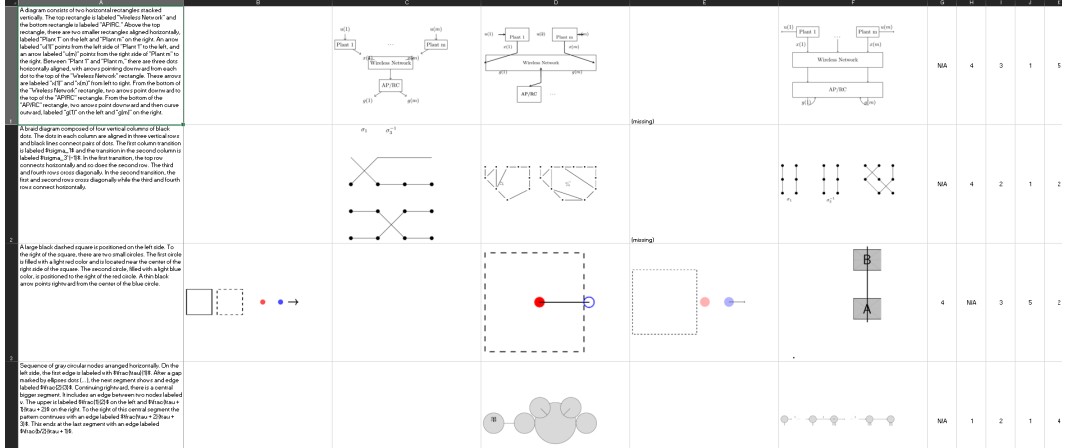

Figure 12: Example of text-image annotations.

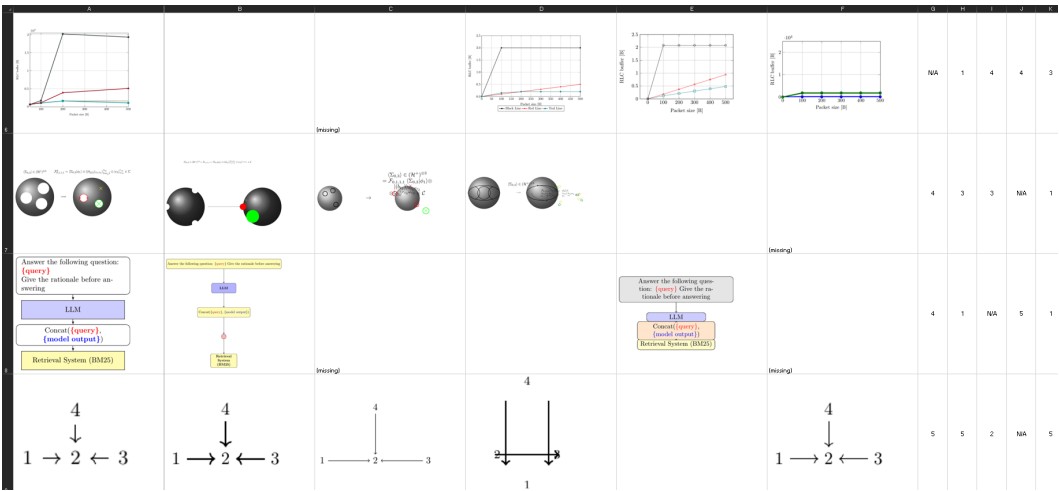

Figure 13: Example of image-image annotations.

(SFT$_{desc. \lor cap.}$), using the full DaTikZ-V4 dataset, but preferring captions whenever they exist, and (iv) Qwen2.5-3B (SFT$_{desc. + cap.}$), oversampling by including both descriptions and captions for all samples with paired captions. This setup isolates whether captions add robustness or simply dilute supervision from richer descriptions.

Table 9 shows that arXiv data alone achieves strong results (0.305 AVG). Adding GitHub yields further gains (0.320), while TeX SE and synthetic data provide marginal benefits. This highlights that large-scale, naturally occurring TikZ from arXiv and GitHub are the most valuable sources.

Table 9: Ablation study of different data sources. Using only data from arXiv already leads to very good performances and arXiV + GitHub almost reaches its full potential.

| Source | CLIP↑ | DSim↑ | TED↓ | AVG↑ | CR↑ | AT |
|---|---|---|---|---|---|---|
| arXiv | 0.152 | 0.568 | 0.805 | 0.305 | 84% | 550 |
| + GitHub | 0.158 | 0.605 | 0.802 | 0.320 | 88% | 548 |
| + TeX SE | 0.159 | 0.608 | 0.806 | 0.320 | 88% | 569 |
| All | 0.161 | 0.613 | 0.802 | 0.324 | 89% | 529 |

FAILURE CASES    We conducted a manual inspection of randomly sampled outputs to compare the typical error patterns of TikZilla and GPT-5. Our observations reveal several systematic differences:

- **Compilation.** GPT-5 frequently produces TikZ code that fails to compile for complex scientific figures, often due to missing library imports, incorrect macro nesting, or hallucinated commands. In contrast, our RL-tuned TikZilla models almost always generate syntactically valid code.

- **Code structure.** TikZilla generally relies on basic primitives such as `\node`, `\draw`, and `\fill`, leading to code that is easy to interpret and modify. GPT-5 tends to generate more elaborate constructs (e.g., macros, loops, nested coordinate definitions), which are more compact but also more brittle and error-prone.

- **Category-specific strengths.** TikZilla performs best on diagrams with strong geometric or mathematical constraints—charts, function plots, schematics, and commutative diagrams (`tikzcd`). GPT-5 performs better on high-level conceptual figures and network style diagrams where spatial layout is loosely specified.

- **Effect of RL tuning.** TikZilla without RL exhibits similar structural strengths but more frequent spatial misalignments (e.g., misplaced labels or arrows). RL substantially improves geometric coherence and spatial consistency.

**Captions vs. Descriptions**

**Caption:** Outline of our algorithm for enumerating Williamson sequences of order n. The boxes on the left correspond to the preprocessing which encodes and decomposes the original problem into SAT instances. The boxes on the right correspond to an SMT-like setup where the system that computes the discrete Fourier transform takes on the role of the theory solver.

**Description:** A block diagram illustrating with several components. There are four main labeled rectangular blocks connected by arrows indicating the direction. At the bottom left, there is an input labeled n entering a rectangular block titled 'Driver script', which sends an arrow labeled 'External call' upward to a block titled 'Diophantine solver / Fourier transform'. From this block another arrow labeled 'Result' points downwards back to the 'Driver script'. From the 'Driver script' a horizontal black arrow point to the right and is labeled 'SAT instances' connected to a block titled 'Programmatic SAT solver'. It outputs a horizontal black arrow labeled 'Enumeration in order n' pointing to the right out of the diagram. Above the 'Programmatic SAT solver' is another block labeled 'Fourier transform' and connected with an upward arrow labeled 'Partial assignment' and a downward arrow labeled 'Conflict clause'. A dashed arrow labeled 'Encoding information' points from the 'Driver script' block back to the ~~'Diophantine solver / Fourier transform'~~ to the 'Fourier transform'.

| Ground Truth | Caption (GPT-4o) | Description (GPT-4o) |
|---|---|---|

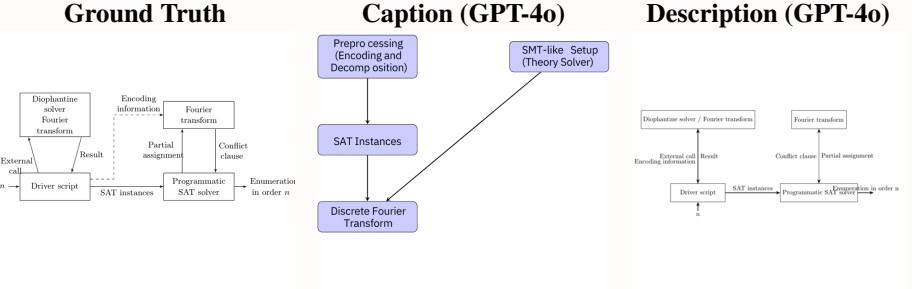

**Caption:** A set $\sigma \in PW+$ inside a rectangle R. The blue region $\frac{R}{(\sigma \cup \partial R)}$ can always be triangulated.

**Description:** A blue rectangle labeled R in the top-left corner. Inside the rectangle, there are two black geometric figures. At the lower-left side, is a layered square pattern composed of three squares, a small black square at the center, surrounded by a blue square matching the background color of the rectangle, surrounded by a larger black square. Diagonally toward the upper-right is an irregular black polygon labeled $\sigma$. Inside the polygon two shapes have the ~~black~~ background color of the rectangle, one is hexagonal at the top and the other is diamond shaped at the bottom.

| Ground Truth | Caption (GPT-4o) | Description (GPT-4o) |
|---|---|---|

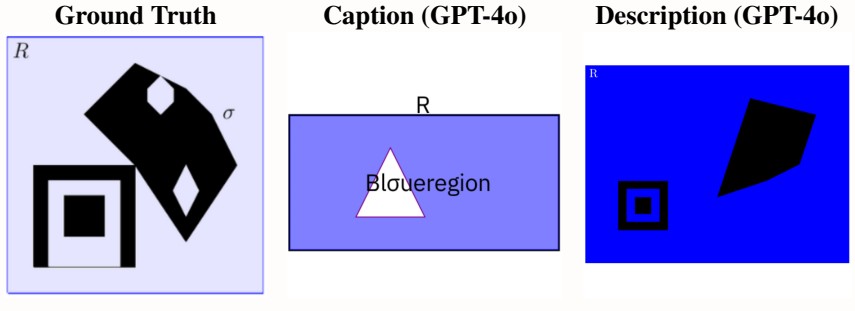

Figure 14: Captions versus VLM-generated descriptions and their produced figures by GPT-4o. ~~Striked out~~ text indicates incorrect VLM-descriptions and red text indicates improved descriptions made by human annotators. We observe that VLMs most frequently omit low-level stylistic details (e.g., small decorative elements), while hallucinations or omission of key structural elements occur less frequently.

## Examples with TikZ Code

**Description & GT Image:**

A large circle centered around the point (0,0). Two points labeled $z^*$ and $w^*$ are placed on the circle near the top. A straight black horizontal line connects both points and is labeled $\theta$ and $2 \sin\left(\frac{\theta}{2}\right)$ and its also exactly aligned with the vertical axes of the circle. From each point $z^*$ and $w^*$ a straight solid line extends downward to the center of the circle. These lines form an isosceles triangle with the lines labeled 1. A vertical dashed line, which is the vertical axes of both the triangle and circle goes from the point (0,0) upwards to the circles circumference and is labeled $\theta$.

**GPT-5:**

```
\documentclass[tikz]{standalone}
\usepackage{amsmath}
\usetikzlibrary{calc}
\begin{document}
\begin{tikzpicture}[scale=1]
\def\R{3}
\def\thetadeg{60}
\coordinate (O) at (0,0);
\coordinate (Z) at ({\R*cos(90-\thetadeg/2)},{\R*sin(90-\thetadeg/2)});
\coordinate (W) at ({\R*cos(90+\thetadeg/2)},{\R*sin(90+\thetadeg/2)});
\coordinate (M) at ($(Z)!0.5!(W)$);
\draw[thick] (O) circle (\R);
\draw[dashed] (O) -- (0,\R) node[midway, right] {$\theta$};
\draw[thick] (O) -- (Z) node[midway, sloped, above] {$1$};
\draw[thick] (O) -- (W) node[midway, sloped, above] {$1$};
\draw[thick] (Z) -- (W);
\node[above] at (M) {$\theta$};
\node[below] at (M) {$2\sin\!\left(\frac{\theta}{2}\right)$};
\fill (Z) circle (2pt) node[above right] {$z^{*}$};
\fill (W) circle (2pt) node[above left] {$w^{*}$};
\end{tikzpicture}
\end{document}
```

**TikZilla-8B-RL:**

```
\documentclass[tikz]{standalone}
\usepackage{tikz}
\usepackage{pgfplots}
\usetikzlibrary{arrows.meta, decorations.pathreplacing, positioning, shapes, calc}
\begin{document}
\begin{tikzpicture}
\draw (0,0) circle (3cm);
\fill (0,0) circle (0.05cm) node[below left] {$0$};
\fill (1.5, 2.598) circle (0.05cm) node[above right] {$z^*$};
\fill (-1.5, 2.598) circle (0.05cm) node[above left] {$w^*$};
\draw[thick] (0,0) -- (1.5, 2.598) node[midway, above left] {$1$};
\draw[thick] (0,0) -- (-1.5, 2.598) node[midway, above right] {$1$};
\draw[dashed] (0,0) -- (0, 3) node[midway, right] {$\theta$};
\node at (0.8, 3.2) {$2\sin\left(\frac{\theta}{2}\right)$};
\end{tikzpicture}
\end{document}
```

Figure 15: TikZ code and rendered figures shown for GPT-5 and TikZilla-3B-RL using the description above. The code produced by GPT-5 more accurately follows the description compared to TikZilla-3B-RL.

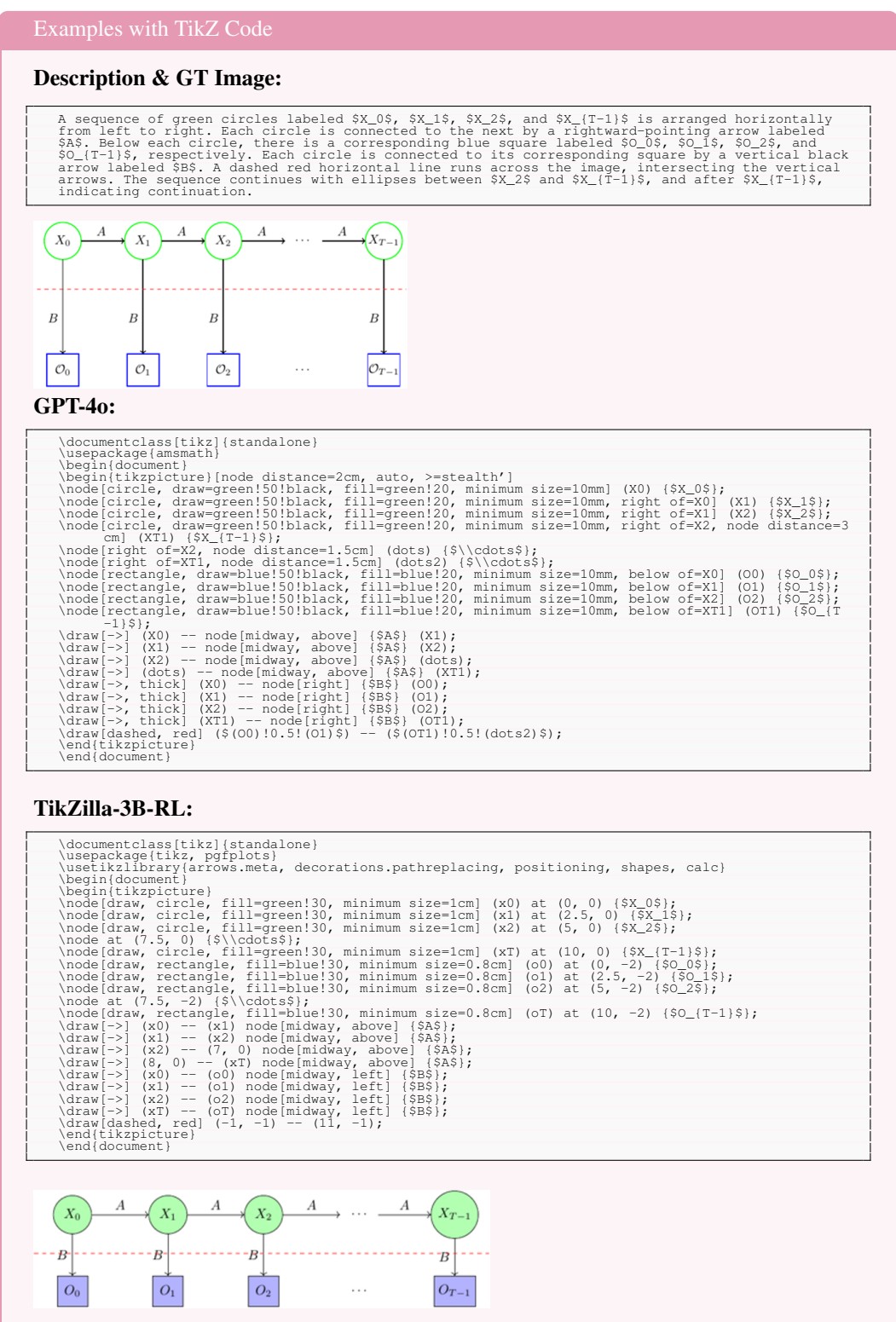

Figure 16: TikZ code and rendered figures shown for GPT-4o and TikZilla-3B-RL using the description above. The code produced by GPT-4o could not be compiled while TikZilla-3B-RL produced correct code.

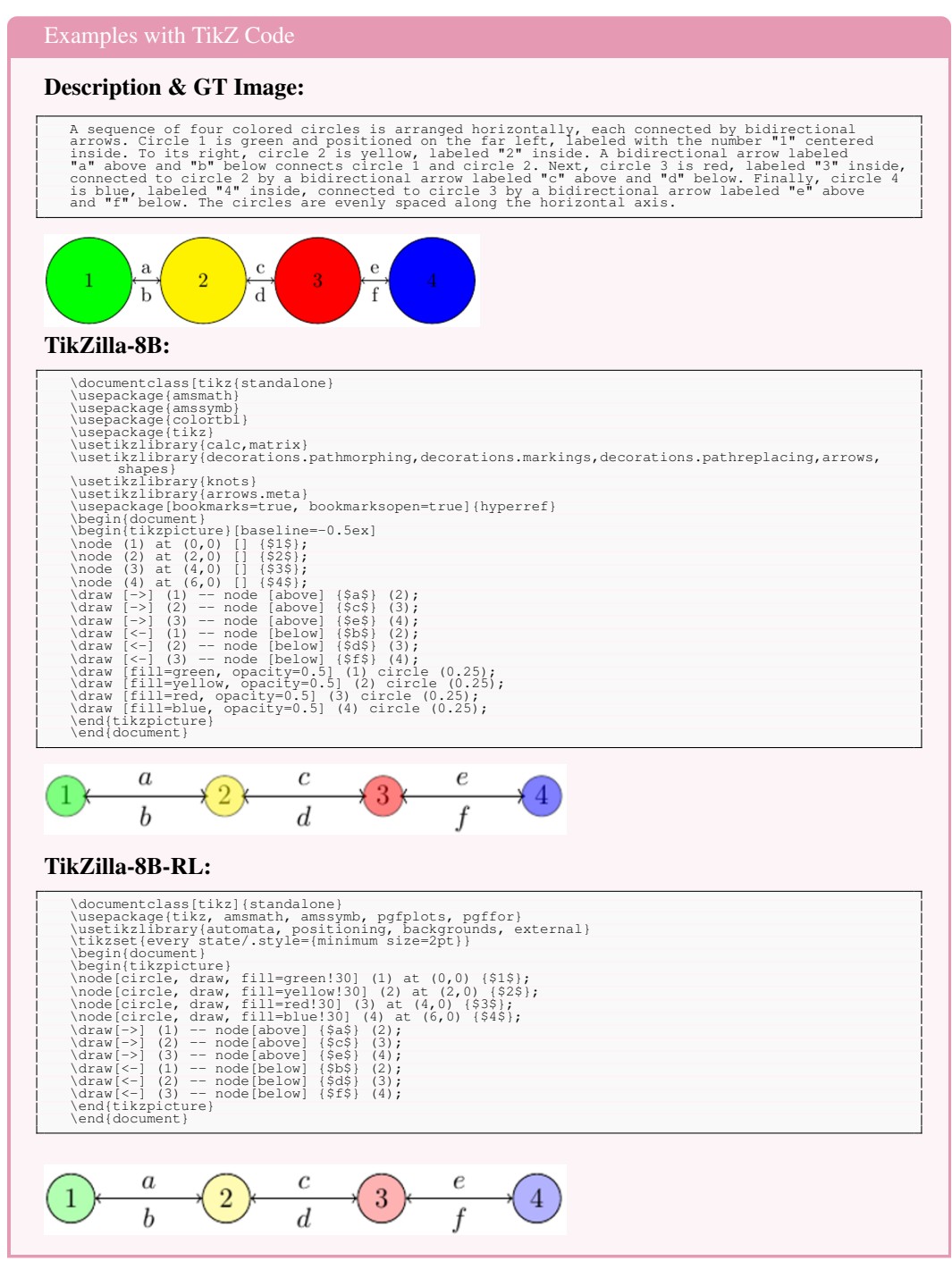

Figure 17: TikZ code and rendered figures shown for TikZilla-8B and TikZilla-8B-RL using the description above. Despite both figures being correct, the code produced by TikZilla-8B-RL is much shorter compared to TikZilla-8B.

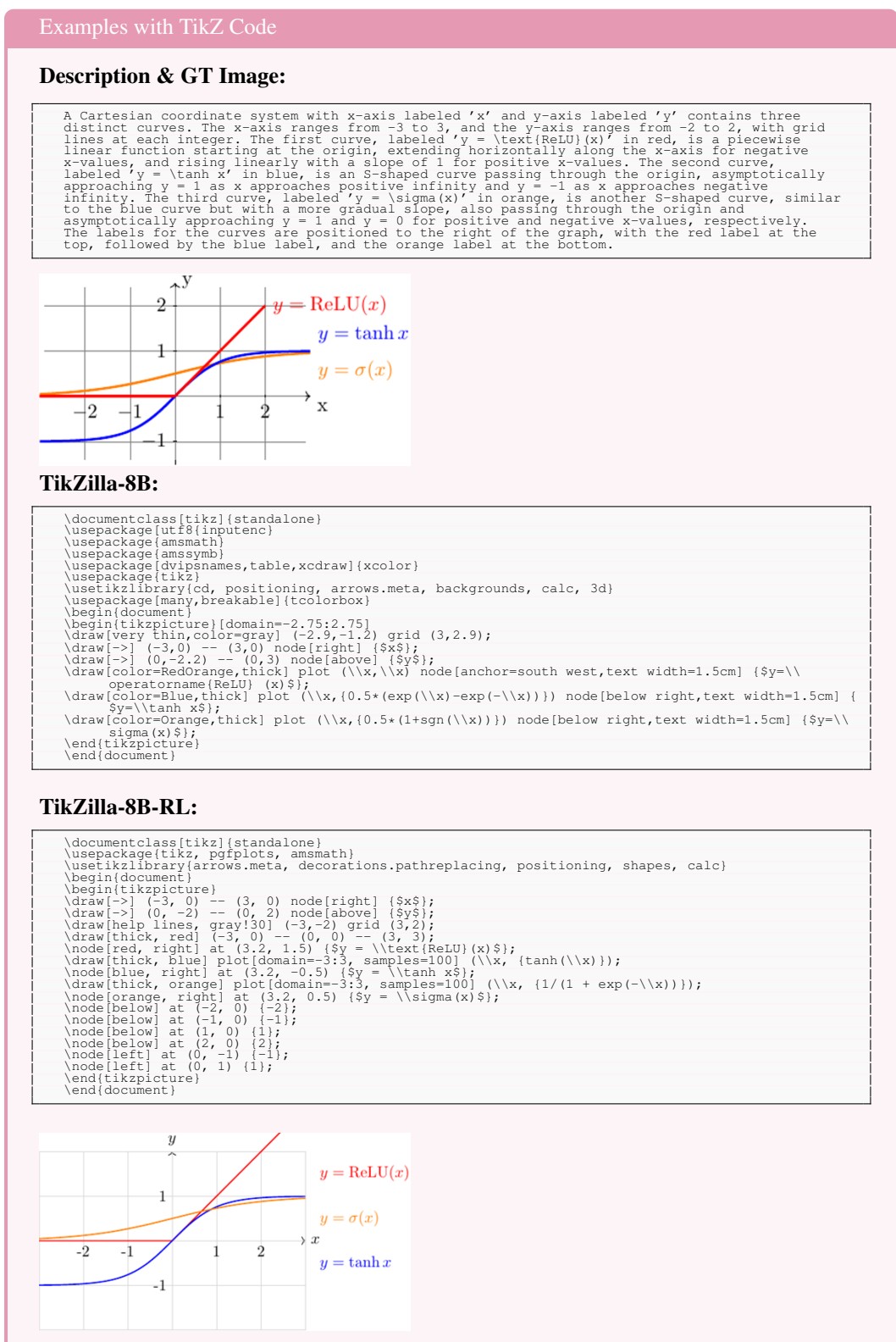

Figure 18: TikZ code and rendered figures shown for TikZilla-8B and TikZilla-8B-RL using the description above. The code produced by TikZilla-8B could not be compiled while TikZilla-8B-RL produced correct code.

Table 10: Exemplary scientific TikZ figures produced by one baseline LLM (GPT-4o) and two of our finetuned LLMs (TikZilla-3B and TikZilla-3B-RL) using the prompts from the first column which have been VLM augmented based on the Ground Truth figures in the second column. ▣-boxed figures have been rated as very good, ▣ as good, ▣ as bad, and ▣ as very bad by human annotators. Empty cells indicate non-compilable TikZ code.

| Prompt | Ground Truth | GPT-4o | TikZilla-3B | TikZilla-3B-RL |
|---|---|---|---|---|
| A series of black lines connect two vertical columns of elements. The left column contains labels $x\_1$, $x\_2$, $x\_3$, $x\_4$, and $x\_n$, arranged vertically from top to bottom with equal spacing. The right column contains shaded rectangles labeled $z\_1$, $z\_2$, $z\_3$, and $z\_m$, also arranged vertically from top to bottom with equal spacing. Each label in the left column is connected by straight black lines to multiple rectangles in the right column, forming a network of intersecting lines. Dotted ellipses are placed vertically between $x\_4$ and $x\_n$ and between $z\_3$ and $z\_m$, indicating continuation. The labels $x\_1$, $x\_2$, $x\_3$, $x\_4$, and $x\_n$ are positioned to the left of their respective lines, while the labels $z\_1$, $z\_2$, $z\_3$, and $z\_m$ are centered within their rectangles. | | | | |
| The bar chart displays accuracy percentages on the y-axis ranging from 80% to 100% with increments of 10%, labeled "Accuracy (%)" on the left. The x-axis is labeled "Number of talkers" and includes five categories: 0, 1, 2, 3, and 4. Each category contains three vertical bars. The first bar is black, representing "MPVAD-SC," the second bar is blue, representing "MPVAD-MC," and the third bar is red, representing "MPVAD-F." Above each bar, there is a numerical label indicating the exact accuracy percentage. For category 0, the black bar is labeled 80, the blue bar 82, and the red bar 85. For category 1, the black bar is labeled 81, the blue bar 83, and the red bar 86. For category 2, the black bar is labeled 82, the blue bar 84, and the red bar 87. For category 3, the black bar is labeled 83, the blue bar 85, and the red bar 88. For category 4, the black bar is labeled 84, the blue bar 86, and the red bar 89. A legend is positioned at the top right corner of the chart, indicating the color and label for each bar type. The chart background includes horizontal dashed lines at each 10% increment on the y-axis. | | | | |
| A diagram consists of several labeled arrows and nodes arranged in a structured format. At the top left, node $\Gamma\_i$ is connected by a rightward arrow labeled $\vdash P$ to node $\Xi\_i$. From $\Xi\_i$, a rightward arrow labeled $\vdash Q$ leads to node $\Psi\_i$. Below $\Gamma\_i$, node $\exists\_i m$ is connected by a downward arrow to node $\Gamma$. From $\Gamma$, a rightward arrow labeled $\exists\_i l$ leads to a central node marked with a circle containing a plus sign. This central node is connected by a rightward arrow labeled $\exists\_j l$ to node $\exists\_j n$. From $\exists\_i n$, a rightward arrow labeled $\Delta$ leads to node $\Psi\_j$. Below $\Gamma$, node $\exists\_j m$ is connected by a downward arrow to node $\Gamma\_j$. From $\Gamma\_j$, a rightward arrow labeled $\vdash P[j/i]$ leads to node $\Xi\_j$. From $\Xi\_j$, a rightward arrow labeled $\vdash Q[j/i]$ leads to node $\Psi\_j$. A dotted arrow labeled $\exists\_i$ connects $\Xi\_i$ to the central node, and another dotted arrow labeled $\exists\_j$ connects the central node to $\Xi\_j$. A vertical arrow labeled $\exists\_i n$ connects $\Psi\_i$ to $\exists\_j n$, and a vertical arrow labeled $\exists\_j n$ connects $\exists\_j n$ to $\Psi\_j$. A horizontal dotted arrow labeled $\Delta\_i = \ell\,z$ connects $\exists\_j n$ to $\Psi\_j$. | | | | |
| A control system diagram features a horizontal line starting from the left with a label $r(t)$, leading to a summation circle. The summation circle has a minus sign on the left and is labeled $e^o(t)$ on the right. From the summation circle, a horizontal line extends rightward into a dashed blue rectangle labeled $C(\alpha\,)$ at the bottom right. Inside the rectangle, there are three vertically aligned blocks labeled $C(\theta\,\_1)$, $C(\theta\,\_k)$, and $C(\theta\,\_N)$ from top to bottom. Each block has a horizontal line extending rightward to a corresponding triangular amplifier labeled $\alpha\,\_1$, $\alpha\,l\,k$, and $\alpha\,\_N$. The outputs of these amplifiers converge at a summation circle on the right side of the rectangle. From this circle, a horizontal line labeled $u(t)$ extends rightward to a block labeled $G$. A horizontal line continues from $G$ to the right, labeled $y^o(t)$. A feedback line loops from $y^o(t)$ back to the summation circle, completing the system. | | | | |
| A rectangular diagram is enclosed by a dashed border with rounded corners. Inside, there are two main vertical paths. The left path begins with a downward arrow labeled "Cond$(\tilde{N}, \tilde{T})$" leading to a rectangle labeled "Up/Down$(\tilde{N}, \tilde{T})$". Below, another downward arrow connects to a rectangle labeled "Conv$\_K(N, T)$", followed by another downward arrow leading to a rectangle labeled "LeakyReLU(0.2)". The right path starts with a downward arrow labeled "Input$(N, T)$" leading to a rectangle labeled "ChannelNorm". Below, a downward arrow connects to a circle with a dot inside, representing a multiplication operation. The left path has a rightward arrow from "Conv$\_K(N, T)$" connecting to a rectangle labeled "Conv$\_K$". This rectangle has a rightward arrow leading to the multiplication circle on the right path. Below the multiplication circle, a downward arrow leads to a circle with a plus inside, representing an addition operation. The left path continues with a rightward arrow from "LeakyReLU(0.2)" connecting to another rectangle labeled "Conv$\_K$". This rectangle has a rightward arrow leading to the addition circle on the right path. Below the addition circle, a downward arrow leads to a label "Output$(N, T)$". The entire diagram is divided into two sections by a vertical dashed line, with the left section containing the "Cond$(\tilde{N}, \tilde{T})$" path and the right section containing the "Input$(N, T)$" path. | | | | |
| A black irregular polygon labeled $(P)\_K$ is centered in the image. Seven black arrows labeled $(U\_K)\_{\sigma\,\_1}$ through $(U\_K)\_{\sigma\,\_7}$ point outward from each vertex of the polygon, with labels positioned near the arrowheads. To the right of the polygon, a set of equations is displayed in black text. The equations are vertically aligned and read as follows: $\mathcal{F}\_K = \{\sigma\,\_i\}\_{i=1}^7$, $\mathcal{U}^{ext}\_K = \{(U\_K)\_{\sigma\,\_i}\}\_{i=1}^7$, $\mathcal{T}\_K = \{\kappa\,\_i\}\_{i=1}^7$, $\mathcal{F}[^\text{ext}]\_K = \{\sigma\,\_i\}\_{i=1}^7$, and $\mathcal{F}\_h = \mathcal{F}[^\text{ext}]\_K \cup \mathcal{F}[^\text{int}]\_K$. The text is right-aligned and positioned to the right of the polygon. | | | | |
| State diagram with two circles labeled $q\_0$ and $q\_2$. Circle $q\_0$ is on the left, connected to circle $q\_2$ on the right by a horizontal arrow labeled "true | int saved = 0; int $x'\_0, \ldots,$ int $x'\_n$; | B" with the label "init" above the arrow. Circle $q\_2$ has a loop arrow on its right side labeled "cond | assert(\pi\,); op; | A" with the label "loop_head" above the loop. Below the diagram, a blue rectangular box contains two lines of text. The first line reads "op $\equiv \text{if}(\text{nondet}()) \wedge \text{saved} = 0)\{x'\_0 = x\_0; \ldots; x'\_n = x\_n; \text{saved} = 1;\}$" and the second line reads "$\pi \equiv (\text{saved} = 1) \implies (x'\_0 \neq x\_0 \lor x'\_1 \neq x\_1 \lor \cdots \lor x'\_n \neq x\_n)$". | | | | |

Table 11: Exemplary scientific TikZ figures produced by one baseline LLM (GPT-5) and two of our finetuned LLMs (TikZilla-8B, and TikZilla-8B-RL) using the prompts from the first column which have been VLM augmented based on the Ground Truth figures in the second column. ■-boxed figures have been rated as very good, ■ as good, ■ as bad, and ■ as very bad by human annotators. Empty cells indicate non-compilable TikZ code.

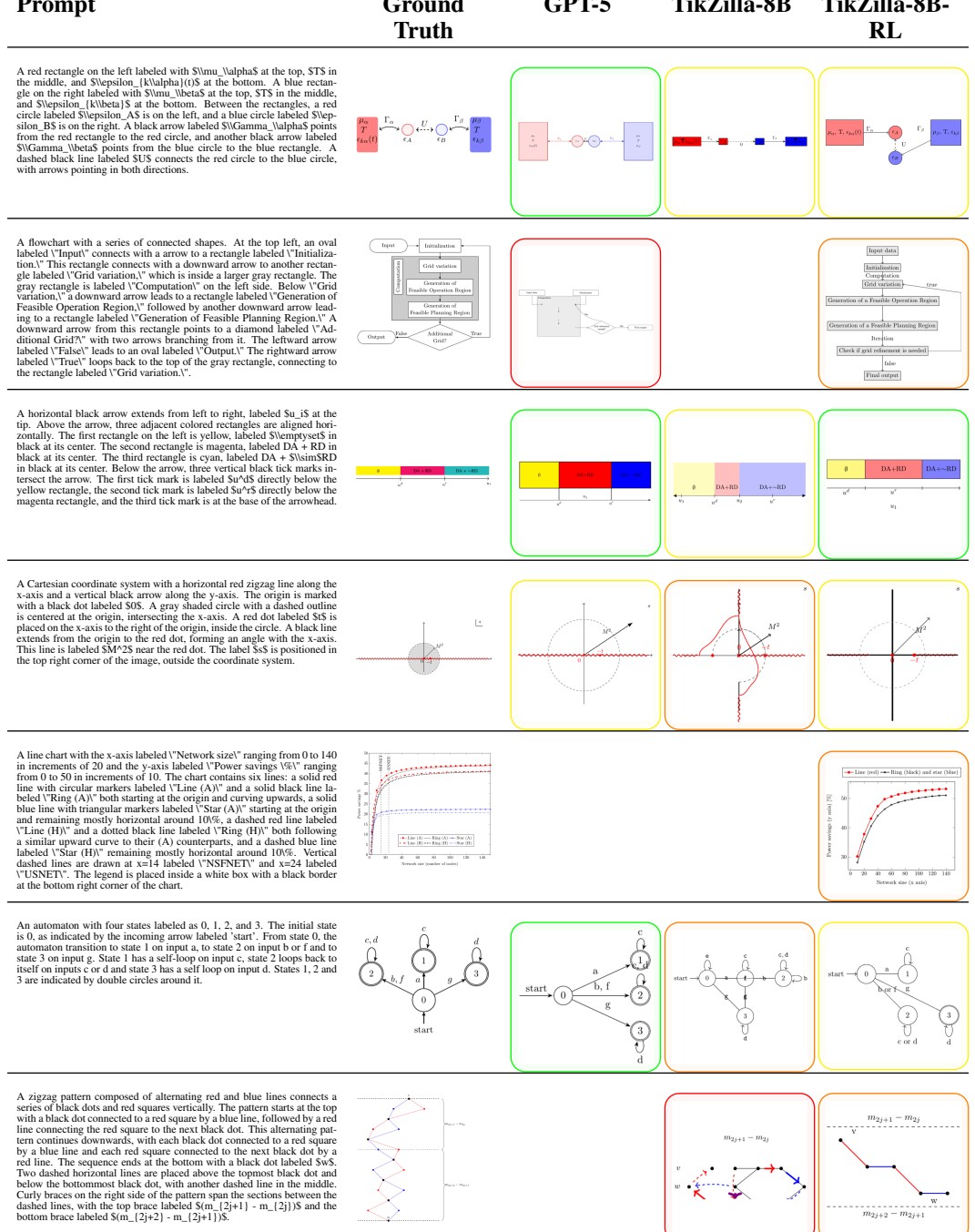

Table 12: Exemplary scientific TikZ figures produced by one baseline LLM (GPT-4o) and two of our finetuned LLMs (TikZilla-8B and TikZilla-8B-RL) using the prompts from the first column which have been VLM augmented based on the Ground Truth figures in the second column. ■-boxed figures have been rated as very good, ■ as good, ■ as bad, and ■ as very bad by human annotators. Empty cells indicate non-compilable TikZ code.

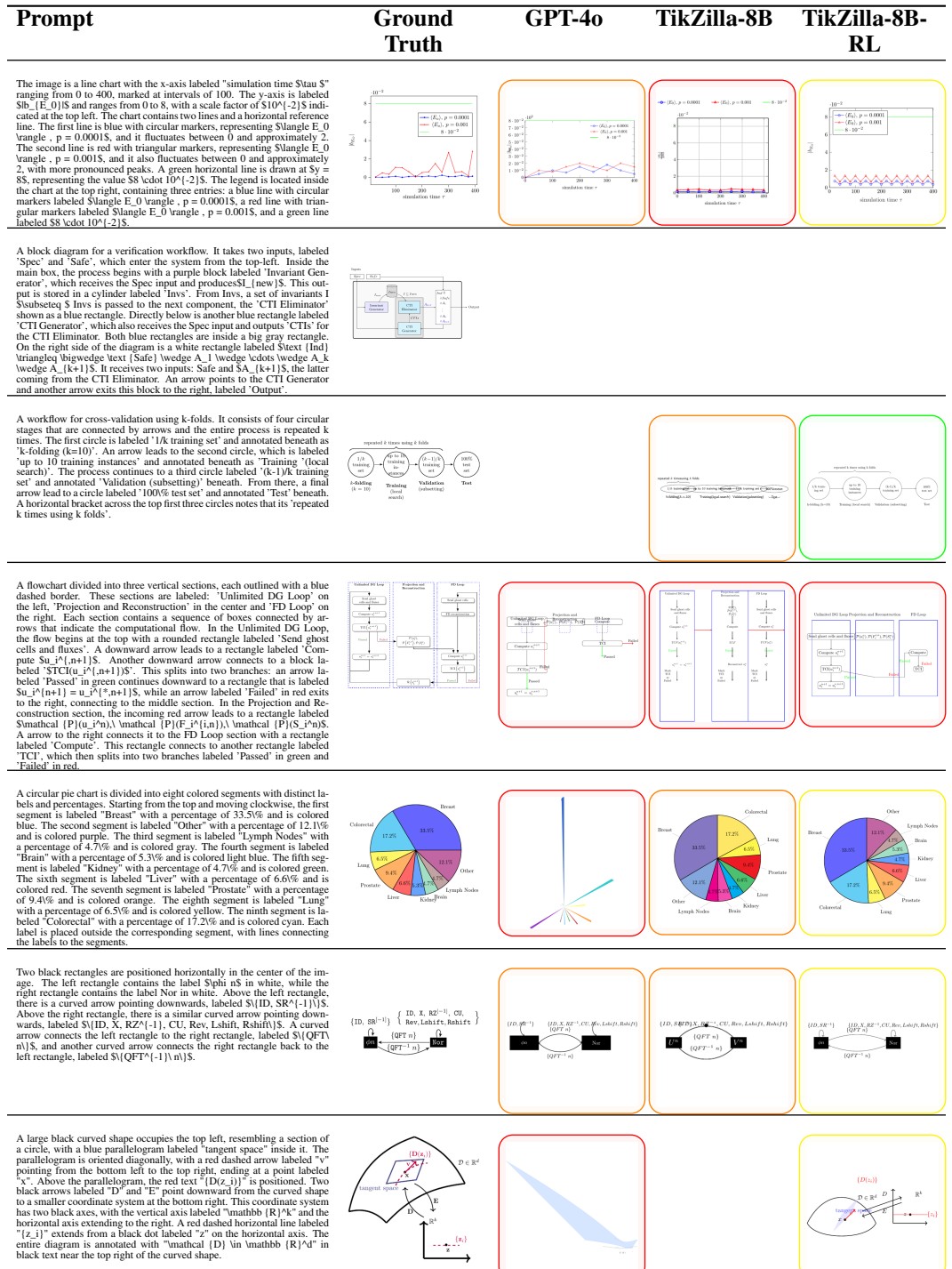

Table 13: Exemplary scientific TikZ figures produced by one baseline LLM (GPT-4o) and two of our finetuned LLMs (TikZilla-8B, and TikZilla-8B-RL) using the prompts from the first column which have been VLM augmented based on the Ground Truth figures in the second column. ■-boxed figures have been rated as very good, ■ as good, ■ as bad, and ■ as very bad by human annotators. Empty cells indicate non-compilable TikZ code.

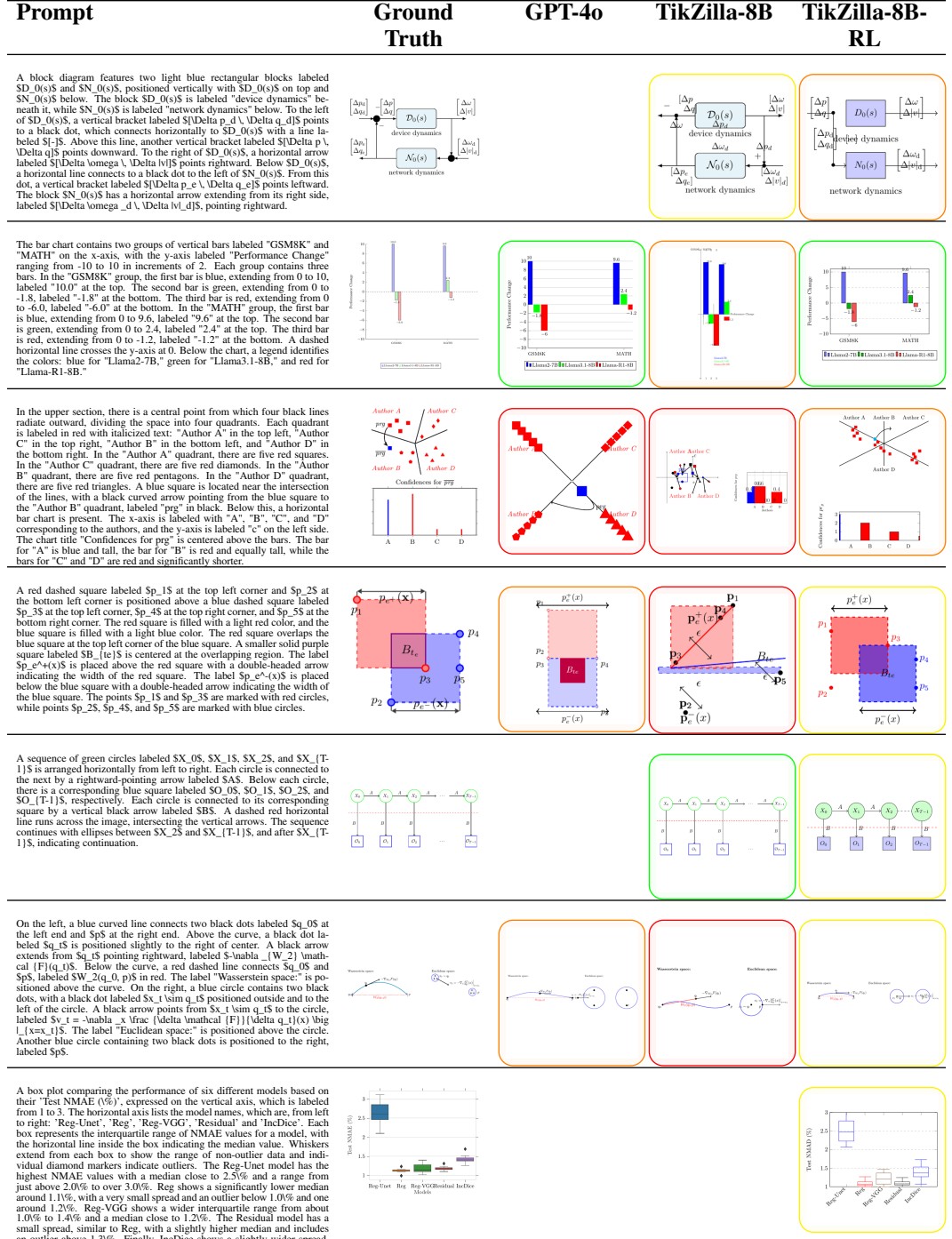

