# OpenReview forum: "TikZilla: Scaling Text-to-TikZ with High-Quality Data and Reinforcement Learning"
_ICLR.cc/2026/Conference — ICLR 2026 Poster_

### Official Review · Reviewer_fGdx · 2025-10-22

**Soundness:** 4
**Presentation:** 4
**Contribution:** 4
**Rating:** 6
**Confidence:** 5

**Summary:**

This paper presents TikZilla, a new family of open-source models for generating scientific figures from text by producing TikZ code. The authors' primary contributions are twofold: 1) the creation of a large-scale, high-quality dataset named DaTikZ-V4, and 2) a two-stage training pipeline. The dataset involves an LLM-based pipeline to debug and repair uncompilable code and a VLM-based process to generate rich figure descriptions, overcoming the sparseness of typical captions. The training methodology first uses SFT to teach the model TikZ syntax, followed by RL to align the output with the desired visual semantics. A key aspect of the RL stage is a domain-specific reward model, derived from a fine-tuned inverse-graphics encoder, which proves to be more effective than generic multimodal metrics. Through extensive automatic and human evaluations, the paper demonstrates that even small TikZilla models (3B/8B) achieve SOTA results, outperforming strong proprietary baselines like GPT-4o and matching the performance of next-generation models on image-based evaluations.

**Strengths:**

This is a strong paper with several key strengths that make it a compelling contribution to the field.

1.  The paper's most significant contribution is the creation of DaTikZ-V4, a large-scale, high-quality dataset for the Text-to-TikZ task. The authors have addressed the critical problem of data scarcity and quality through a meticulous, multi-source collection process and two highly innovative enhancement steps: (1) an LLM-based debugging pipeline to repair uncompilable code, substantially increasing the usable data, and (2) the use of VLMs to generate semantically rich descriptions, which are demonstrably superior to the sparse original captions. The appendix further validates the data quality with detailed analysis.

2.  The proposed domain-specific reward model for RL is a standout feature. By finetuning an inverse-graphics (Image-to-TikZ) model and using the semantic distance between image embeddings as a reward signal, the authors have developed a method that is simple, intuitive, and highly effective. This approach is more semantically faithful for scientific figures than general-purpose metrics, and its strong correlation with human judgment validates its design and utility in guiding the model towards generating visually accurate figures.

3.  The paper presents a solid two-stage training pipeline (SFT followed by RL) that is executed with rigor, evidenced by the detailed description of the multi-GPU, long-duration training process. The authors clearly articulate the role and limitations of each stage, providing a strong justification for their approach. For instance, the paper notes that while SFT trains the model in syntax, it "remains unaware of the rendered semantics," which perfectly motivates the necessity of the subsequent RL stage for visual alignment.

4. The experimental evaluation is extensive and highly convincing. The authors benchmark their models against a wide range of the latest and most powerful systems on a carefully constructed, contamination-free test set. The results are outstanding: TikZilla achieves state-of-the-art performance across multiple metrics, significantly outperforming top open-source models and even much larger proprietary systems like GPT-4o. Crucially, these strong quantitative results are corroborated by a thorough human evaluation with expert annotators, confirming that TikZilla produces publication-quality figures.

5.  The paper is exceptionally well-written and clearly structured. The motivation is compelling, the problem is well-defined, and the proposed solutions are presented in a logical, easy-to-follow narrative. The authors excel at explaining the rationale behind each design choice, making the entire research story coherent and persuasive.

**Weaknesses:**

1. The entire training pipeline is fundamentally dependent on the quality of the VLM-generated figure descriptions. While the paper demonstrates that these are superior to raw captions, VLMs are known to hallucinate and omit critical details.
2. The paper convincingly argues for its domain-specific reward model (R_sim) by showing its high correlation with human judgment. However, it doesn't contain a direct experimental comparison against general-purpose reward models (like CLIPScore or DreamSIM) within the RL training loop.
3. The reward model operates on the semantic similarity of image patch embeddings. It may not be sensitive enough to fine-grained stylistic details that are crucial for scientific figures, such as precise line weights, specific dash patterns, exact color hex codes, or font styles.
4. The dataset is sourced primarily from arXiv, GitHub, and TeX StackExchange, which likely results in a corpus rich in diagrams, plots, and graphs common to fields like computer science, mathematics, and physics. But it's unclear how well TikZilla would generalize to more structurally different (OOD) types of scientific figures, such as complex chemical reaction pathways, intricate biological diagrams, or detailed engineering schematics, which may be underrepresented in the training data.

**Questions:**

1. Could you provide a more quantitative analysis of the VLM's error rate? For example, by manually evaluating a random sample of generated descriptions against their ground-truth images, could you estimate how often the VLM omits key details, hallucinates elements, or misinterprets spatial relationships?
2. While I understand that a full training run with a different reward model is resource-intensive, could you provide more direct evidence of $R_{sim}$'s practical superiority?
3. Have you observed any limitations regarding the model's ability to render specific styles, such as different dash patterns, line weights, or precise color shades, when these are explicitly requested in the prompt? Since the reward is based on patch embeddings, it might not penalize deviations in these subtle attributes as strongly.
4. Table 3 shows an interesting result: RL applied directly to the base Qwen3-8B model (+RL) yields a significant performance boost (AVG 0.251 → 0.357), whereas for Qwen2.5-3B, the improvement is much smaller. The paper suggests this is because larger models already encode some TikZ knowledge. Could you elaborate on this hypothesis? Is the primary role of SFT for smaller models simply to get them into a "syntactically plausible" region where RL can effectively take over?

---

> ### Author Response · Authors · 2025-11-22
>
> We sincerely appreciate the reviewers' positive feedback on our paper.
>
> > The entire training pipeline is fundamentally dependent on the quality of the VLM-generated figure descriptions. While the paper demonstrates that these are superior to raw captions, VLMs are known to hallucinate and omit critical details.
>
> > Could you provide a more quantitative analysis of the VLM's error rate? For example, by manually evaluating a random sample of generated descriptions against their ground-truth images, could you estimate how often the VLM omits key details, hallucinates elements, or misinterprets spatial relationships?
>
> Appendix Fig. 13 contains a qualitative error analysis in which human experts annotate VLM-generated descriptions for omissions (struck-through text) and hallucinations (red text). In a pilot evaluation of this type, we observe that VLMs most frequently omit low-level stylistic details (e.g., minor ticks, precise labels, small decorative elements), while hallucinations or omission of key structural elements occur less frequently for typical scientific figures. However, for highly complex or domain-specialized diagrams, VLMs do sometimes produce briefer descriptions in which important details are missing. While fully human-annotated descriptions would be ideal, they are infeasible even at a smaller scale. We view our approach as a practical and scalable solution that benefits from ongoing improvements in state-of-the-art VLMs.
>
> > The paper convincingly argues for its domain-specific reward model (R_sim) by showing its high correlation with human judgment. However, it doesn't contain a direct experimental comparison against general-purpose reward models (like CLIPScore or DreamSIM) within the RL training loop.
>
> > While I understand that a full training run with a different reward model is resource-intensive, could you provide more direct evidence of R_{sim}'s practical superiority?
>
> We thank the reviewer for this suggestion. We completed two new experiments directly comparing our domain-specific reward $R_{sim}$​ against CLIPScore (image–image) and DreamSIM, performed on the supervised-finetuned Qwen2.5-3B model using identical GRPO hyperparameters. Since CLIPScore and DreamSIM appear in some of our evaluation metrics, we report performance on two independent metrics (DINOScore and LPIPS [1]) to avoid reward–metric coupling.
>
> | **LLM**                                                | **DINO ↑** | **LPIPS ↑** | **TED ↓** | **AVG ↑** | **CR ↑** | **AT** |
> |--------------------------------------------------------|------------|-------------|-----------|-----------|----------|--------|
> | Qwen2.5-3B $(+SFT + RL_{CLIP_{Img}})$                       | 0.751      | 0.418       | 0.779     | 0.463     | 97%      | 537    |
> | Qwen2.5-3B $(+SFT + RL_{DSim})$                           | 0.759      | 0.439       | 0.777     | 0.474     | 99%      | 494    |
> | Qwen2.5-3B $(+SFT + RL_{R_{Sim}(DaTikZ-V3)})$         | 0.789      | 0.440       | 0.768     | 0.487     | 97%      | 496    |
> | TikZilla-3B-RL                                         | 0.809      | 0.451       | 0.766     | 0.498     | 98%      | 481    |
>
> The results indicate that all reward functions improve over the SFT baseline, but our $R_{sim}$ consistently achieves the strongest AVG performance. DreamSIM performs slightly better than CLIPScore, which is consistent with our correlation analysis with human judgments. These findings provide direct empirical evidence that a domain-specific reward model tailored to scientific diagrams is more effective in our RL setting than general-purpose perceptual similarity models. We will add this to the updated version.
>
> [1]: Zhang, Richard, et al. "The unreasonable effectiveness of deep features as a perceptual metric." Proceedings of the IEEE conference on computer vision and pattern recognition. 2018.

---

> > ### Author Response · Authors · 2025-11-22
> >
> > > The reward model operates on the semantic similarity of image patch embeddings. It may not be sensitive enough to fine-grained stylistic details that are crucial for scientific figures, such as precise line weights, specific dash patterns, exact color hex codes, or font styles.
> >
> > > Have you observed any limitations regarding the model's ability to render specific styles, such as different dash patterns, line weights, or precise color shades, when these are explicitly requested in the prompt? Since the reward is based on patch embeddings, it might not penalize deviations in these subtle attributes as strongly.
> >
> > This is an excellent point, and we agree that patch-based image embeddings can, in principle, miss very fine stylistic attributes. In practice, however, we find that our reward model is more sensitive to such details than one might expect. This is because the image encoder used in the reward is trained as part of DeTikZify, an inverse-graphics model that must reconstruct full TikZ code from rendered scientific figures. To do this successfully, the encoder must capture subtle cues such as line weights, dash patterns, arrow styles, and even font variations, otherwise the decoder cannot reproduce the correct TikZ primitives. Empirically, we observe that modifying a single fine-grained detail (e.g., switching from dashed to densely dashed, adjusting line width, or slightly altering color) consistently leads to a measurable decrease in the reward score, indicating that the patch embeddings encode these attributes to a meaningful extent.
> >
> > > The dataset is sourced primarily from arXiv, GitHub, and TeX StackExchange, which likely results in a corpus rich in diagrams, plots, and graphs common to fields like computer science, mathematics, and physics. But it's unclear how well TikZilla would generalize to more structurally different (OOD) types of scientific figures, such as complex chemical reaction pathways, intricate biological diagrams, or detailed engineering schematics, which may be underrepresented in the training data.
> >
> > We agree with the reviewer that evaluating TikZilla on OOD data is important. To this end, we use the SPIQA [2] dataset to benchmark our model beyond the TikZ-centric DaTikZ-V4 corpus. SPIQA figures are typically not generated in TikZ but instead originate from matplotlib/ggplot2/MATLAB pipelines or consist of multi-panel scientific figures, overlays, and varied diagrammatic layouts. This makes SPIQA a meaningful OOD test from a rendering-style and structural-complexity perspective. We used all samples from the test-A and test-B splits and generated textual descriptions using GPT-4o, yielding 397 test cases. Since ground-truth TikZ code is unavailable, we omit TED in this evaluation.
> >
> > | **LLM**                   | **CLIP ↑** | **DSim ↑** | **CR ↑** | **AT** |
> > |---------------------------|------------|-------------|----------|--------|
> > | GPT-5                     | 0.115      | 0.432       | 60%      | 1239   |
> > | GPT-4o                    | 0.098      | 0.326       | 48%      | 748    |
> > | Qwen3-Coder-30B-A3B   | 0.102      | 0.349       | 58%      | 1000   |
> > | Qwen2.5-3B                | 0.038      | 0.117       | 24%      | 772    |
> > | TikZilla-3B               | 0.114      | 0.374       | 64%      | 1170   |
> > | TikZilla-3B-RL        | **0.193**  | **0.637**   | **97%**  | 765    |
> > | Qwen3-8B                  | 0.070      | 0.228       | 37%      | 781    |
> > | TikZilla-8B               | 0.131      | 0.428       | 70%      | 1184   |
> > | TikZilla-8B-RL            | *0.174*    | *0.584*     | *90%*    | 809    |
> >
> > Relative to the DaTikZ-V4 test split (Table 3  in the paper), SPIQA exhibits significantly longer sequences, lower compilation rates, and overall lower performance, as expected due to its non-TikZ origins and higher visual complexity. Intriguingly, both TikZilla-8B-RL and especially TikZilla-3B-RL outperform GPT-5 on this OOD benchmark.
> >
> > [2]: Pramanick, Shraman, Rama Chellappa, and Subhashini Venugopalan. "Spiqa: A dataset for multimodal question answering on scientific papers." Advances in Neural Information Processing Systems 37 (2024): 118807-118833.

---

> > > ### Author Response · Authors · 2025-11-22
> > >
> > > > Table 3 shows an interesting result: RL applied directly to the base Qwen3-8B model (+RL) yields a significant performance boost (AVG 0.251 → 0.357), whereas for Qwen2.5-3B, the improvement is much smaller. The paper suggests this is because larger models already encode some TikZ knowledge. Could you elaborate on this hypothesis? Is the primary role of SFT for smaller models simply to get them into a "syntactically plausible" region where RL can effectively take over?
> > >
> > > Yes, the reviewer’s interpretation is correct. For smaller models such as Qwen2.5-3B, the base model lacks strong TikZ competence. Its generations often fail to compile or are structurally incoherent. In this regime, the main function of SFT is to teach core TikZ syntax and common structural patterns, so that the model enters a region where RL can obtain informative visual rewards at all. Without this, RL receives mostly zero or noisy rewards, limiting its ability to improve the model. For larger models like Qwen3-8B, the base model already emits syntactically valid and roughly structured TikZ code even before SFT. As a result, RL immediately operates in a much more informative reward landscape where it can refine geometry, spatial relations, and semantic fidelity.

---

> ### Author Response · Authors · 2025-11-27
>
> Dear Reviewer fGdx,
>
> thank you again for your positive and valuable review.
> As the author response deadline is approaching, is there anything else we could clarify?
> Else, if you think that our rebuttal addresses your concerns, we would of course be happy if you would increase your score.
>
> Best regards,
> Authors

---

> > ### Comment · Reviewer_fGdx · 2025-11-28
> > **Nice rebuttal**
> >
> > I thank the authors for their detailed and comprehensive response. The rebuttal effectively addresses my primary concerns.
> >
> > Specifically:
> > - Reward Model Comparison: The new experiments comparing R_sim against CLIPScore and DreamSIM (measured on independent metrics like DINOScore and LPIPS) provide the direct empirical evidence I requested. This confirms the practical superiority of the domain-specific reward model for this task.
> > - OOD Generalization: The evaluation on the SPIQA dataset is a significant addition. It demonstrates that TikZilla generalizes well to scientific figures outside its primary training distribution, outperforming strong proprietary baselines even on non-TikZ layouts.
> > - VLM Limitations: The explanation regarding the VLM error analysis and the reference to the qualitative analysis in the Appendix are reasonable.
> >
> > The inclusion of these additional experiments significantly strengthens the paper and solidifies the claims regarding the contribution of DaTikZ-V4 and the training pipeline. Given the high quality of the presentation, the soundness of the approach, and the now-fortified experimental results, I am raising my score to 8.

---

> > > ### Author Response · Authors · 2025-11-29
> > >
> > > Thanks for your thoughtful follow-up. We really appreciate your feedback and are glad the additional experiments adressed your concerns.

---

### Official Review · Reviewer_9Bdk · 2025-10-27

**Soundness:** 3
**Presentation:** 3
**Contribution:** 3
**Rating:** 6
**Confidence:** 3

**Summary:**

This paper focuses on advancing the state-of-the-art in the application of text-to-Tikz, the challenge of generating code for a tikz diagram based on a text specification. The paper provides several contributions. The first is an analysis of existing text-to-Tikz datasets. The paper finds that relying on captions for the textual description of a figure, as many of these datasets apparently do, is probably insufficient for the development of strong models since captions fail to capture major structural components. To mitigate this, the paper introduces DaTikZ-V4, which revises and extends a previous dataset called DaTikZ-V3. This includes more careful filtering, additional data sources like GitHub, and the inclusion of LVLM figure descriptions. After this, the paper describes its approach to training text-to-TikZ models using supervised finetuning and then RL with a special emphasis on the RL rewards used. Finally, the paper describes a range of comparisons between models trained with the proposed methods (the paper calls these TikZilla models) and other relevant LLMs, showing that Tikzilla models are competitive with much larger, proprietary systems.

**Strengths:**

**Clarity:** The paper is well-written. It effectively communicates its contributions relative to other work in this area so that even a reader that is not familiar with the text-to-TikZ problem can appreciate the results. DaTikZ-V4 is well-motivated by an analysis of existing datasets (though the reviewer would have appreciated a few more examples). The figures are all of high quality and efficiently visualize key ideas in the paper.

**Effective, small, specialized models:** One potential criticism of this paper is that the TikZilla models are at best marginally better than many existing models (e.g., GPT5). The important point though (from this reviewer’s perspective) is that the TikZilla models are much, much smaller than GPT5 and other proprietary models that achieve comparable performance. This reviewer believes that the generation of small, reliable, specialized models is an important contribution to the community.

**Weaknesses:**

**A lot of the paper feels routine:** The approach described in the paper appears to result in substantially better performance than prior small-model approaches. However, the methods used to achieve these improvements are exactly what most readers would likely expect: increase the size of the dataset, increase the quality of the dataset, implement recent approaches that have been successful in language modeling broadly (e.g., RL). One exception to this is the section on reward signals specific for the text-to-TikZ problem. Together, the reviewer believes that this will somewhat limit what a reader will take away from the work. On the other hand, a result being in line with community intuition should not be a barrier to publication. As such, the reviewer did not weight this ‘weakness’ very strongly when evaluating the work.

**More analysis of failure cases would be interesting:** The paper devotes considerable space to describing how current text-to-TikZ datasets fall short of what would likely be needed to train strong text-to-TikZ models. The reviewer appreciated this contribution. It would have been nice to see a similar type of analysis applied to the results section where text-to-TikZ models were compared. TikZilla and GPT5 score comparably in certain respects, but this doesn’t mean that they fail in the same ways. If large proprietary models and TikZilla have different strengths and weaknesses, this would be very important to know in practice.

**Nitpicks**
- Line 268: A parenthesis is incorrectly oriented.
- Line 365: This would probably come across as more interesting if it was not labeled as ‘emergence’, which feels like a strong word for this particular phenomenon.
- Table 3: It would be helpful to remind the reader what the acronyms stand for so they don’t have to flip back to earlier sections.

**Questions:**

-	Line 478: Why is this approach “ethical”?
-	Will models or datasets be released?

---

> ### Author Response · Authors · 2025-11-22
>
> We are happy to receive the reviewer’s feedback.
>
> > A lot of the paper feels routine: The approach described in the paper appears to result in substantially better performance than prior small-model approaches. However, the methods used to achieve these improvements are exactly what most readers would likely expect: increase the size of the dataset, increase the quality of the dataset, implement recent approaches that have been successful in language modeling broadly (e.g., RL). One exception to this is the section on reward signals specific for the text-to-TikZ problem. Together, the reviewer believes that this will somewhat limit what a reader will take away from the work. On the other hand, a result being in line with community intuition should not be a barrier to publication. As such, the reviewer did not weight this ‘weakness’ very strongly when evaluating the work.
>
> We appreciate the comment and understand that parts of our pipeline may appear intuitive in isolation. However, our primary contributions go beyond simply “scaling up” or “applying RL”. We found several aspects that, to our knowledge, had not been examined before. In particular, we provide:
>
> (i) a systematic analysis showing that widely available captions are insufficient for figure reconstruction, motivating the need for higher-fidelity supervision.
>
> (ii) a substantially improved dataset with compiler-aware filtering, VLM-based descriptions, and LLM debugging.
>
> (iii) a task-specific reward model based on an inverse-graphics encoder, which yields stronger RL improvements than general-purpose perceptual rewards.
>
> Taken together, these components enable small open models to match or surpass much larger proprietary systems, which we believe is a very useful and non-trivial takeaway for the community.
>
> > More analysis of failure cases would be interesting: The paper devotes considerable space to describing how current text-to-TikZ datasets fall short of what would likely be needed to train strong text-to-TikZ models. The reviewer appreciated this contribution. It would have been nice to see a similar type of analysis applied to the results section where text-to-TikZ models were compared. TikZilla and GPT5 score comparably in certain respects, but this doesn’t mean that they fail in the same ways. If large proprietary models and TikZilla have different strengths and weaknesses, this would be very important to know in practice.
>
> We agree that this is an important point. In our manual inspection of randomly sampled outputs, we observe the following distinctions:
>
> **1.) Compilation:** GPT-5 frequently produces TikZ code that fails to compile for complex scientific images, typically due to missing library imports, incorrect nested macros, or hallucinated commands. In contrast, our RL-tuned TikZilla models almost always produce syntactically valid TikZ code.
>
> **2.) Code style:** TikZilla tends to rely on basic, widely-used primitives such as \node, \draw, and \fill. This results in code that is easy to interpret and modify. GPT-5 often generates more elaborate constructions (e.g., macros, coordinate definitions inside loops), which are more prone to syntax errors and harder to debug when they fail.
>
> **3.) Category-specific:** TikZilla consistently outperforms GPT-5 on diagrams with strong geometric or mathematical structure such as charts, function plots, schematics, and commutative diagrams (tikz-cd). GPT-5 is better for high-level conceptual figures and network-style diagrams where the semantics are more abstract and geometry is less constrained.
>
> **4.) Effect of RL tuning:** TikZilla without RL shares similar strengths but shows more frequent spatial misalignments (e.g., misplaced arrows or labels), whereas RL significantly improves geometric coherence.
>
> We will add a short summary of the observed failure categories to the revised version, in addition to the examples already included in the appendix (Tables 8–11).
>
> > Nitpicks
>
> > Line 268: A parenthesis is incorrectly oriented.
>
> > Line 365: This would probably come across as more interesting if it was not labeled as ‘emergence’, which feels like a strong word for this particular phenomenon.
>
> > Table 3: It would be helpful to remind the reader what the acronyms stand for so they don’t have to flip back to earlier sections.
>
> We thank the reviewer for these suggestions and will incorporate all of them in the revised version.
>
> > Line 478: Why is this approach “ethical”?
>
> Our intention was to highlight that open-source models naturally enable transparency, reproducibility, and auditability, which are often discussed as important ethical aspects of scientific research. We will clarify this in the CR.
>
> > Will models or datasets be released?
>
> Yes. We plan to release all four TikZilla models and the portion of DaTikZ-V4 that is cleared for redistribution on HuggingFace. The full DaTikZ-V4 dataset, as well as all training and inference pipelines, will be released on GitHub.

---

### Official Review · Reviewer_LicD · 2025-10-30

**Soundness:** 3
**Presentation:** 3
**Contribution:** 2
**Rating:** 6
**Confidence:** 3

**Summary:**

This paper proposes a data generation methodology to generate 1.5M tikz examples. These are then used to finetune QWen (with RL + SFT) in order to obtain small-ish (3B/8B) models that surpass GPT-4 and are similar in performance to GPT-5.

The authors do this by first sourcing data from Github and other sources and then using a pipeline to identify valid tikz and leveraging a VLM to obtain descriptions.

This data is used to finetune and then with RL using GRPO.

**Strengths:**

1. This paper describes how to obtain a large scale text-to-tikz dataset. They use a combination of choices (e.g. 1 tikz per website; ensuring compilation; VLM style description) to obtain a large scale and high quality dataset.

2. They describe howt o use such a dataset for SFT and for RL. For RL, they add a couple of domain specific changes -- e.g. the model for the reward; the scalar rewards for capturing semantic alignment.

3. Their strategy seems to work -- leading to clear improvements across model sizes and leading to performance that is comparable with GPT-4o / GPT-5 though much smaller by using a human rating comparison.

**Weaknesses:**

1. It would be good to quantify the difference in model size in Figure 4 -- how much smaller is the model than GPT-5/4o in terms of sheer size of the model as well as compute at inference time.

2. it would be good to plot / understand performance as a function of the size of the data. If we want better performance, can we just collect more data or are we already hitting diminishing returns here ?

3. Is the dataset planning to be made public? It would be useful for other researchers I imagine.

**Questions:**

See above.

---

> ### Author Response · Authors · 2025-11-22
>
> We would like to thank the reviewer for their feedback.
>
> > It would be good to quantify the difference in model size in Figure 4 -- how much smaller is the model than GPT-5/4o in terms of sheer size of the model as well as compute at inference time.
>
> Since GPT-4o and GPT-5 are proprietary models, their exact parameter counts and inference FLOPs have not been publicly disclosed. However, public expert estimates place these models in the range of 1.8 trillion parameters. In contrast, our TikZilla models are 3B and 8B parameters respectively. While we cannot compare exact inference compute due to the absence of published FLOPs for GPT-4o/GPT-5, it is reasonable to expect that TikZilla requires orders of magnitude less compute at inference due to its much smaller model size and absence of mixture-of-experts architectures.
>
> > it would be good to plot / understand performance as a function of the size of the data. If we want better performance, can we just collect more data or are we already hitting diminishing returns here ?
>
> We agree that understanding performance as a function of dataset size is important. To study this, we trained Qwen2.5-3B (SFT) on subsets of DaTikZ-V4 at 75%, 50%, 25%, 12.5%, and 6.25% of the full dataset size.
> | **LLM**                     | **CLIP ↑** | **DSim ↑** | **TED ↓** | **AVG ↑** | **CR ↑** | **AT** |
> |-----------------------------|------------|------------|-----------|-----------|----------|--------|
> | TikZilla-3B                 | 0.161      | 0.613      | 0.802     | 0.324     | 89%      | 672    |
> | TikZilla-3B (75%)       | 0.148      | 0.564      | 0.803     | 0.303      | 85%      | 704    |
> | TikZilla-3B (50%)       | 0.133      | 0.520      | 0.808     | 0.282      | 76%      | 769    |
> | TikZilla-3B (25%)       | 0.132      | 0.510      | 0.816     | 0.275      | 78%      | 841    |
> | TikZilla-3B (12.5%)    | 0.122      | 0.465      | 0.813      | 0.258     | 71%      | 775    |
> | TikZilla-3B (6.25%)    | 0.104      | 0.386      | 0.821      | 0.223     | 61%      | 847    |
> | Qwen2.5-3B               | 0.081      | 0.315      | 0.789      | 0.202     | 52%      | 387    |
>
> Results (AVG score) show a monotonic improvement across all metrics as training data increases, with no indication of diminishing returns within the explored range. This suggests that further data scaling (e.g., synthetic data) is a promising direction for future work. We will include that in the CR version.
>
> > Is the dataset planning to be made public? It would be useful for other researchers I imagine.
>
> Yes. We will release the portion of DaTikZ-V4 whose licenses permit redistribution (primarily GitHub and permissively licensed arXiv sources) on HuggingFace. For the full dataset, including content that cannot be redistributed directly, we will release a GitHub repository containing the complete data-generation pipeline, enabling researchers to fully reconstruct DaTikZ-V4 from scratch.

---

### Official Review · Reviewer_XPus · 2025-11-01

**Soundness:** 2
**Presentation:** 3
**Contribution:** 3
**Rating:** 6
**Confidence:** 4

**Summary:**

This paper presents DaTikZ-V4, a new large-scale dataset designed for text-to-TikZ generation.
In addition to the conventional supervised fine-tuning (SFT) approach, the authors introduce a reinforcement learning framework that explicitly takes into account the visual quality of the generated images.
By employing a two-stage training strategy that combines large-scale data with reinforcement learning, the proposed method achieves a substantial improvement in text-to-TikZ generation performance.

**Strengths:**

- The authors newly construct a large-scale TikZ dataset, approximately four times larger than the existing dataset.
- They define a reward function based on image similarity and demonstrate the effectiveness of reinforcement learning.
- They conduct not only automatic evaluations but also human evaluations, demonstrating the effectiveness of the proposed method.

**Weaknesses:**

- In Section 4, the paper describes the collection of a new large-scale dataset; however, there appears to be no mention of its license information. Since license details are essential for enabling data reuse, it would be helpful to provide not only the total number of data samples but also a breakdown of the dataset by license type.
- The correlation between the automatic evaluation metrics and the human evaluation results appears to be low, making it difficult to accurately assess the quality of the generated results based solely on the automatic scores. For example, in Table 4, the discussion is based only on the automatic evaluation values. It would therefore be desirable to develop or employ automatic evaluation metrics that exhibit a higher correlation with human judgments.

**Questions:**

In Section 3, under Caption Quality Analysis, the paper states that “Accurate text-to-TikZ generation requires captions that specify objects, attributes, and spatial relations.”
While it is understandable that detailed captions are helpful for reducing hallucinations during model training, in practical applications it is often costly to describe every element in an image exhaustively. In many real-world scenarios, users may prefer to generate diagrams from shorter or more abstract textual instructions rather than from fully detailed descriptions.
Given that the proposed model is trained only on detailed captions, how does it perform when provided with insufficiently detailed or abstract instruction texts? What kind of outputs would it produce in such cases?

---

> ### Author Response · Authors · 2025-11-22
>
> We appreciate the reviewers’ time and feedback and would like to address the perceived weaknesses and questions.
>
> > In Section 4, the paper describes the collection of a new large-scale dataset; however, there appears to be no mention of its license information. Since license details are essential for enabling data reuse, it would be helpful to provide not only the total number of data samples but also a breakdown of the dataset by license type.
>
> Our analysis suggests that 35.55% of all snippets originate from papers under Creative Commons licenses, which permit redistribution (e.g., CC-BY, CC-BY-SA, CC0). The remaining 64.45% originate from papers using the arXiv Nonexclusive-Distribution License or papers without an explicit license, which do not permit third-party redistribution. A more fine-grained breakdown is as follows:
>
> - arXiv Nonexclusive-Distribution License: 40.03%
>
> - No license / unknown: 24.43%
>
> - CC-BY 4.0: 25.52%
>
> - CC-BY-NC-ND 4.0: 4.86%
>
> - CC-BY-NC-SA 4.0: 2.87%
>
> - CC-BY-SA 4.0: 1.56%
>
> - CC0: 0.73%
>
>
> > The correlation between the automatic evaluation metrics and the human evaluation results appears to be low, making it difficult to accurately assess the quality of the generated results based solely on the automatic scores. For example, in Table 4, the discussion is based only on the automatic evaluation values. It would therefore be desirable to develop or employ automatic evaluation metrics that exhibit a higher correlation with human judgments.
>
> We agree with the reviewer that designing a dedicated task-specific automatic metric is important. However, we consider this to be beyond the scope of the current paper (p.9, L482–484). At the same time, not all automatic metrics correlate poorly with human judgments in our setting. In particular, DreamSIM shows a correlation of 0.586 with human preferences (p.8, L427), which we consider moderate–strong. Nevertheless, because automatic metrics may not fully capture fine-grained semantic fidelity in complex TikZ figures, we complemented them with an extensive human evaluation consisting of over 1,000 expert judgments, which we believe provides a reliable assessment of model quality.
>
> > In Section 3, under Caption Quality Analysis, the paper states that “Accurate text-to-TikZ generation requires captions that specify objects, attributes, and spatial relations.” While it is understandable that detailed captions are helpful for reducing hallucinations during model training, in practical applications it is often costly to describe every element in an image exhaustively. In many real-world scenarios, users may prefer to generate diagrams from shorter or more abstract textual instructions rather than from fully detailed descriptions. Given that the proposed model is trained only on detailed captions, how does it perform when provided with insufficiently detailed or abstract instruction texts? What kind of outputs would it produce in such cases?
>
> To test this, we manually evaluated TikZilla on (i) short, moderately abstract prompts, (ii) underspecified prompts, and (iii) highly abstract prompts. For moderately abstract instructions (e.g., “a bar chart with three categories”, “a commutative diagram with two arrows”), TikZilla produces correct TikZ code (e.g., placing three bars of different heights or drawing a node-arrow configuration). For underspecified prompts (e.g., “plot a function”, “draw a diagram with nodes and arrows”), the model also produces correct TikZ code (e.g., plotting a placeholder mathematical curve/producing a minimal node-edge structure). However, for highly abstract descriptions (e.g., “The figure presents a question and answer format…”), the model often embeds the text directly into the figure. This behavior is expected, as these prompts do not contain object-, geometry-, or relation-level instructions that our TikZilla model can translate into drawing primitives.

---

### Author Response · Authors · 2025-12-01
**Summary Comment to AC**

We again thank the reviewers for their time and constructive feedback and believe that in our rebuttal, we have addressed all concerns raised. Specifically we included the following information in our uploaded manuscript:

- Dataset licenses (Reviewer 1): We added a full license breakdown (CC, nonexclusive, unknown) and clarified what fraction can be redistributed (Page 4, lines 188-191).

- Dataset scale (Reviewer 2): We ablate performance as a function of the size of the data. Results show consistent improvements up to 100% of DaTikZ-V4, with no saturation observed (Page 9-10, lines 482-497).

- Failure case analysis (Reviewer 3): We compared typical error patterns of GPT-5 and our TikZilla models (Appendix Page 24, lines 1251-1268).

- VLM description quality (Reviewer 4): We added human–human baselines, demonstrating VLM generated descriptions are semantically close to human-written ones (Page 3, line 119; Page 4, lines 162-163). Additionally, we provided estimations to VLMs errors (Appendix Page 25, lines 1347-1349).

- Reward models (Reviewer 4): We added an ablation comparing our domain-specific reward R_{sim} with CLIP_{Img} and DreamSIM (Page 9, lines 445-453, 475-478). We also included two additional metrics (DinoScore and LPIPS) to avoid reward-metric coupling (Page 7, lines 344-348).

- Out-of-distribution evaluation (Reviewer 4): We conducted an analysis using data from SPIQA to show how our TikZilla models generalizes to non-TikZ scientific diagrams. Results show that both of our TikZilla models outperform all baseline models including GPT-5 (Page 9, lines 498-520).

We also appreciate that reviewer 4 indicated during the discussion phase that our rebuttal fully resolved their concerns. It is a pity that we could not follow-up with the remaining reviewers, but we are confident that they are happy with our revisions and our rebuttal. We hope this summary is helpful for the AC, and we are happy to provide any further clarification if needed.

---

### Meta-Review · Area_Chair_tmrY · 2026-01-04

**Summary:**

This paper introduces a higher-quality Text-to-TikZ dataset and shows that combining supervised fine-tuning with an RL stage can substantially improve figure generation, even allowing small open models to match or surpass much larger proprietary systems. The most notable technical contribution is the use of a domain-specific, image-based reward that aligns generated TikZ code with rendered visual semantics.

Reviewers’ concerns mainly focused on the incremental nature of the contribution, noting that the overall approach follows an expected recipe of better data, better filtering, and RL without introducing a fundamentally new idea. Rs also pointed out the heavy reliance on automatically generated captions and learned rewards, which may miss fine details or limit robustness outside the dominant figure types in the dataset. Finally, Rs and AC both are both concerned about the paper's generalization and reuse for future scope, observing that while the RL setup is effective for Text-to-TikZ, it is task-specific and nontrivial to reproduce, which tempers the broader impact of the work. AC suggests accept but poster level is reasonable

**Reviewer Concerns:**

Across reviewers, the main recurring concern was that the paper’s core contribution is largely incremental. Several reviewers noted that the overall recipe is well aligned with current community expectations anyway. While the execution is okay and the results are competitive, especially for small open models, the work does not introduce a fundamentally new modeling paradigm, unfortunately. This limits the conceptual novelty, even though the outcome is somewhat practically useful.

The overall theme depends on automatically generated supervision and rewards. Reviewers questioned how robust the approach is to errors in VLM-generated descriptions and whether the learned visual reward truly captures all aspects of figure quality, especially fine-grained stylistic details. The rebuttal attempted to address this with additional analyses and ablations, which alleviated some concerns, but reviewers and AC would still view this reliance as an inherent limitation of the approach rather than a fully resolved issue if the discussion phase is not interrupted.

Several reviewers raised questions about generalization and long-term impact. Although the authors added out-of-distribution evaluations and data-scaling ablations, reviewers remained cautious about how broadly the conclusions extend beyond TikZ-style scientific figures and about how reusable the RL setup is for other domains. The RL component was consistently seen as interesting and well motivated, but also task-specific and non-trivial to reproduce.

**Reviewer Scores:**

fGdx raises score to 8.

Others remain.

So final: 6 6 6 8

---

### Decision · Program_Chairs · 2026-01-26

Accept (Poster)